# 🔺 `PRISM`: A 3D Probabilistic Neural Representation for Interpretable Shape Modeling

Yining Jiao[1]   Sreekalyani Bhamidi[1]   Carlton Jude Zdanski[2]   Julia S Kimbell[2]   Andrew Prince[2]
Cameron P Worden[2]   Samuel Kirse[2]   Christopher Rutter[2]   Benjamin H Shields[2]   William Alexander Dunn[2]
Jisan Mahmud[1]   Marc Niethammer[3]

## Abstract

Understanding how anatomical shapes evolve in response to developmental covariates—and quantifying their spatially varying uncertainties—is critical in healthcare research. Existing approaches typically rely on global time-warping formulations that ignore spatially heterogeneous dynamics. We introduce `PRISM`, a novel framework that bridges implicit neural representations with uncertainty-aware statistical shape analysis. `PRISM` models the conditional distribution of shapes given covariates, providing spatially continuous estimates of both the population mean and covariate-dependent uncertainty at arbitrary locations. A key theoretical contribution is a closed-form Fisher Information metric that enables efficient, analytically tractable local temporal uncertainty quantification via automatic differentiation. Experiments on three synthetic datasets and one clinical dataset demonstrate `PRISM`'s strong performance across diverse tasks—from modeling shape evolution to personalized shape prediction and anomaly detection—within a unified framework, while providing interpretable and clinically meaningful uncertainty estimates. The code is publicly available at https://github.com/uncbiag/PRISM.

[1]Department of Computer Science, University of North Carolina at Chapel Hill, Chapel Hill, USA [2]School of Medicine, University of North Carolina at Chapel Hill, Chapel Hill, USA [3]Department of Computer Science, University of California San Diego, La Jolla, USA. Correspondence to: Yining Jiao <jyn@cs.unc.edu>, Marc Niethammer <mniethammer@ucsd.edu>.

*Proceedings of the 43rd International Conference on Machine Learning*, Seoul, South Korea. PMLR 306, 2026. Copyright 2026 by the author(s).

## 1. Introduction

Statistical shape modeling (SSM) aims to quantify biological variation by capturing the distribution of anatomical geometries. Many healthcare applications require modeling how anatomy changes with continuous covariates such as age $t$ while learning the conditional density $p(\mathcal{Y} \mid t)$ (Bône et al., 2018a; Hong et al., 2017). Critically, even at a fixed $t$, anatomy exhibits substantial inter-subject variability that is spatially heteroscedastic: some regions vary far more than others. Quantifying this spatially varying uncertainty is essential for distinguishing developmentally conserved regions from naturally diverse ones, enabling robust anomaly detection and personalized assessment.

Despite its importance, incorporating rigorous uncertainty quantification into covariate-aware shape modeling remains a challenge. Existing approaches generally fall into two primary categories, each with distinct limitations. On the one hand, `NAISR` achieves high-fidelity conditional generation but is fundamentally deterministic, providing point estimates of deformations without quantifying confidence or population variance. On the other hand, classical statistical atlases (Durrleman et al., 2013; Bône et al., 2018a) explicitly model shape variability using diffeomorphic mapping; yet, they face a critical representation gap. Specifically, while these methods consider variability within a parameter space (e.g., distributions of initial momenta or time-shifts), propagating this uncertainty to the image domain requires integrating velocity fields through non-linear deformations. Consequently, obtaining an explicit, pointwise map of aleatoric uncertainty directly on the anatomy is not analytically tractable and typically relies on computationally intensive numerical resampling (e.g., Monte Carlo strategies). Thus, a significant gap remains for a framework that offers a theoretically grounded, closed-form formulation capable of directly mapping intrinsic biological ambiguity onto the anatomical space for direct shape analysis.

We introduce `PRISM` (Probabilistic Interpretable Shape Modeling), a framework that bridges neural implicit representations with information geometry. `PRISM` models

shape evolution as a continuous heteroscedastic Gaussian field and derives a closed-form Fisher Information metric for analytic uncertainty quantification at arbitrary spatial locations.

Specifically, our contributions are as follows: (1) a conditional probabilistic implicit field that jointly models the mean developmental trajectory and spatially varying population variability from cross-sectional data; (2) a closed-form Fisher Information metric that enables efficient and analytically tractable local uncertainty quantification via automatic differentiation; and (3) an amortized inverse encoder that estimates intrinsic developmental time from local shapes without test-time optimization.

Experiments on synthetic and clinical datasets across diverse tasks—shape evolution, intrinsic time inference, personalized prediction, and OOD detection—demonstrate PRISM's versatility and strong performance compared to state-of-the-art baselines.

## 2. Related Work

**Shape Analysis.** Statistical shape analysis quantifies population variability and covariate effects on anatomy (Dryden & Mardia, 2016; Kendall, 1984; Heimann & Meinzer, 2009). Classical point distribution models (PDMs) summarize variation via PCA (Cootes et al., 1995; Goodall, 1991), but they operate on discrete coordinates and do not provide an analytic, continuous uncertainty field over the anatomical domain. Diffeomorphic registration and atlas frameworks (e.g., LDDMM/Deformetrica) preserve topology and enable population modeling (Beg et al., 2005; Durrleman et al., 2013; Bône et al., 2018b), yet their statistics are typically defined in a tangent parameter space (e.g., initial momenta/velocities), making pointwise uncertainty on the anatomy difficult to obtain in closed form (Zhang et al., 2017; Bône et al., 2018a). Gaussian Process Morphable Models introduce kernel-based probabilistic priors (Lüthi et al., 2017), but their uncertainty is tied to the kernel/observation model and is not naturally conditioned on time/covariates nor designed for incomplete observations. *PRISM bridges this gap by modeling a continuous spatiotemporal field with uncertainty directly conditioned on covariates, enabling spatially resolved uncertainty quantification on anatomy.*

**Uncertainty Estimation.** Predictive uncertainty is commonly decomposed into aleatoric (data-inherent) and epistemic (model) components (Kendall & Gal, 2017; Hüllermeier & Waegeman, 2021; Abdar et al., 2021). In medical imaging, a large body of work approximates epistemic uncertainty using sampling-based methods, e.g., Monte Carlo dropout (Gal & Ghahramani, 2016) and deep ensembles (Lakshminarayanan et al., 2017), supporting tasks such as segmentation (Jungo & Reyes, 2019), re-

construction (Vasconcelos et al.), and visualization (Saklani et al., 2024). In clinical shape analysis, however, the primary operational need is often aleatoric uncertainty (Hüllermeier & Waegeman, 2021): spatially localized, covariate-conditioned population variability that distinguishes physiological variation from pathology. Existing computational anatomy pipelines quantify variability largely in parameter space (e.g., LDDMM momenta) (Durrleman et al., 2013; Bône et al., 2018a; Zhang et al., 2017). *In contrast, PRISM provides an analytically tractable, spatially heteroscedastic uncertainty field on the anatomy that can be queried directly at arbitrary resolution.*

**Neural Representations.** Neural implicit representations model geometry as continuous, resolution-agnostic functions (Sethian, 1999; Osher & Fedkiw, 2005; Park et al., 2019; Mescheder et al., 2019). Neural deformation models further represent dense deformations using implicit fields (Zheng et al., 2021), with medical applications to shape modeling (Yang et al., 2022) and deformable image registration (Wolterink et al., 2022). A related line builds implicit *image* atlases conditioned on covariates, differing in how they absorb inter-subject variability: CINA (Dannecker et al., 2024) and CINeMA (Dannecker et al., 2025) encode each subject in a latent code while avoiding deformation fields, whereas SINA (Großbröhmer et al., 2024) uses per-subject deformations. These operate in the image domain and produce only a *mean* atlas. NAISR (Jiao et al., 2024) conditions neural deformations on covariates but remains deterministic. None provides a covariate-conditioned, spatially heteroscedastic uncertainty field. Dynamic neural representations with time-warping have been used for 4D sequences (Nizamani et al., 2025), yet they typically lack uncertainty quantification and often assume closed, complete surfaces, which limits their applicability to cross-sectional clinical data. *PRISM extends neural implicit shape modeling by estimating spatiotemporal heteroscedastic uncertainty and supporting learning with incomplete observations.*

## 3. Problem Formulation

**Dataset.** We consider a dataset comprising $N$ shape observations $\{\mathcal{Y}_i\}_{i=1}^{N}$, each associated with a corresponding covariate $t_i$ (e.g., time or age). Let $\Omega \subset \mathbb{R}^3$ denote a canonical domain on which we define a fixed reference template $\mathcal{T}$. Since the template shape is shared across the population, each observed shape $\mathcal{Y}_i$ is fully characterized by its deviation from $\mathcal{T}$. We model this deviation via a displacement field $\phi_i : \Omega \to \mathbb{R}^3$, which maps each template coordinate $\boldsymbol{p}$ to a displacement vector $\boldsymbol{d}$. For any point $\boldsymbol{p}$ on the template $\mathcal{T}$, the field $\phi_i(\cdot)$ outputs a 3D displacement $\boldsymbol{d} = \phi_i(\boldsymbol{p}) \in \mathbb{R}^3$, and the corresponding location on the $i$-th observed shape is $\boldsymbol{y} = \boldsymbol{p} + \boldsymbol{d}$. Thus, the $i$-th shape

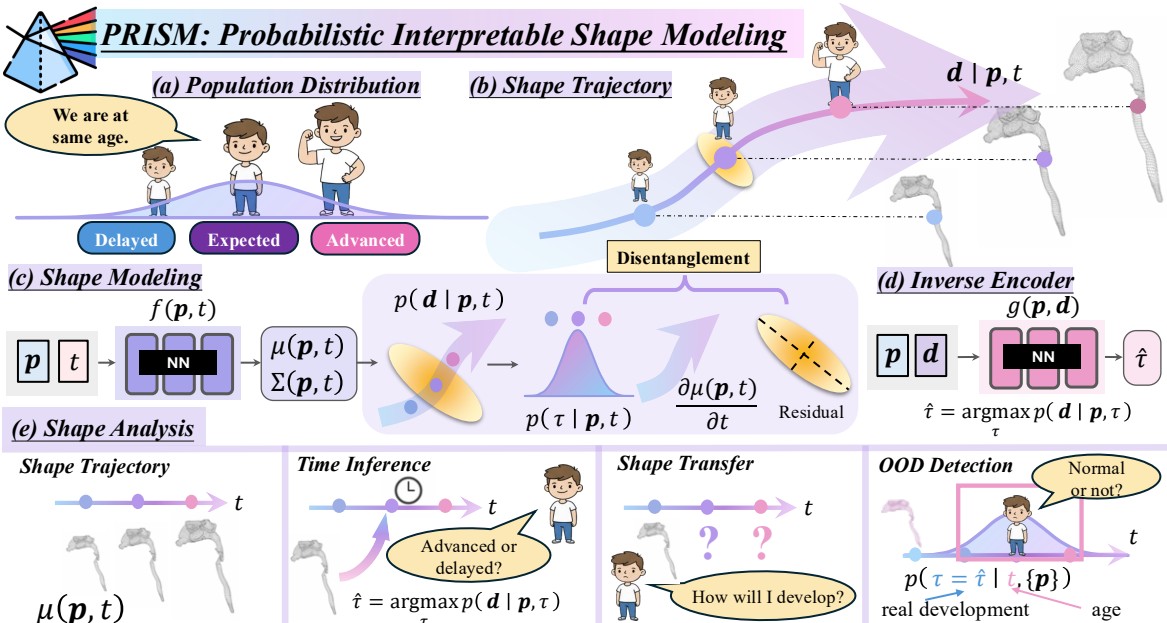

*Figure 1.* Overview of the PRISM framework. (a) illustrates the population distribution of developmental stages (e.g., physiological ages) at a fixed covariate (e.g., chronological age), indicating that individuals can be developmentally delayed, expected, or advanced. (b) visualizes the probabilistic shape deformation trajectory $p(d \mid p, t)$, differentiating between variations in *developmental progression* (indicated by blue, purple, pink dots) and *intrinsic shape attributes* independent of the time $t$ (represented by the orange plate). (c) details the shape modeling component, where a neural network (NN) estimates the mean trajectory $\mu(p, t)$ and total covariance $\Sigma(p, t)$. These learned parameters enable the estimation of temporal uncertainty through the Fisher information (Eq. (12)). (d) presents the Inverse Encoder, which facilitates downstream analysis by inferring the developmental time $\hat{\tau}$ from a template query point $p$ and deformation $d$, serving as the foundation for tasks in (e). (e) shows various applications of PRISM in shape analysis, including generating population-level shape trajectories, inferring individual developmental stage (time inference), predicting future shapes (shape transfer), and detecting abnormal development (OOD detection) by evaluating the likelihood of an observed shape within the population distribution.

is represented as $\mathcal{Y}_i = \{p + \phi_i(p) \mid p \in \mathcal{T}\}$, and shape variability across the population is encoded entirely in the displacement fields $\{\phi_i\}$.

Although our ultimate interest is in the target position $y$, we model the displacement $d$ directly. Since the template point $p$ is fixed, the linear relationship $y = p + d$ implies that the distributions are equivalent up to a deterministic translation:

$$p(y \mid p, t) \equiv p(d{+}p \mid p, t) = p(d \mid p, t) = \mathcal{N}(\mu(p, t), \Sigma(p, t)) \tag{1}$$

Crucially, this translation leaves the covariance structure *invariant* ($\Sigma_y = \Sigma_d$). This invariance allows us to quantify the geometric uncertainty of the target position $y$ directly by modeling the displacement $d$.

**Point Correspondence.** Our framework operates on shapes represented as displacement fields from a common template, requiring point correspondence across subjects—each template coordinate $p$ maps to anatomically equivalent locations across all shapes. This correspondence is established via a template-based registration model that learns bidirectional deformations between a learned shared template and individual anatomies. As this is not our primary contribution, we defer implementation details to Appendix A.1.

**Goal.** We model the conditional distribution of displacement $d$ at template point $p$ given covariate $t$ as $p(d \mid p, t) = \mathcal{N}(\mu(p, t), \Sigma(p, t))$, capturing both the mean trajectory and heteroscedastic variability (Kendall & Gal, 2017). Central to our formulation is *intrinsic time* $\tau$, a latent variable representing true developmental progression distinct from chronological time $t$—subjects at identical $t$ may be at different points along the shape evolution trajectory. We are particularly interested in quantifying this *temporal uncertainty*: the distribution $p(\tau \mid p, t)$ characterizing temporal variability at each spatial location.

Our framework targets three quantities: (1) **Conditional Shape Distribution** $p(d \mid p, t)$, the mean trajectory and total variability; (2) **Individual Intrinsic Time** $\hat{\tau} = g(p, d)$, mapping observed deformation to its developmental stage; (3) **Temporal Uncertainty** $p(\tau \mid p, t)$, quantifying developmental variability at location $p$ and time $t$. A summary of notation is provided in Tab. 6.

# 4. Methodology

Building on the problem formulation in Sec. 3, this section details the computational framework for estimating the three targeted quantities: (1) the conditional shape distribution $p(\boldsymbol{d} \mid \boldsymbol{p}, t)$; (2) the individual intrinsic time $\hat{\tau}$; and (3) the intrinsic time distribution $p(\tau \mid \boldsymbol{p}, t)$.

## 4.1. Conditional Shape Distribution $p(\boldsymbol{d} \mid \boldsymbol{p}, t)$

To capture the population-level trajectory $\mu(\boldsymbol{p}, t)$ and its associated variability $\Sigma(\boldsymbol{p}, t)$, we model the displacement field as a continuous, coordinate-based conditional distribution. We propose a *probabilistic implicit representation*, where the observed displacement $\boldsymbol{d}$ is treated as a random variable conditioned on the chronological time $t$ and the template coordinate $\boldsymbol{p}$.

Formally, we model $p(\boldsymbol{d} \mid \boldsymbol{p}, t)$ as a heteroscedastic Gaussian field,

$$\boldsymbol{d} \mid \boldsymbol{p}, t \sim \mathcal{N}\big(\mu(\boldsymbol{p}, t), \Sigma(\boldsymbol{p}, t)\big), \quad \forall \boldsymbol{p} \in \Omega, \ t \in \mathbb{R}, \quad (2)$$

where $\mu(\boldsymbol{p}, t)$ represents the *mean developmental trajectory*, and $\Sigma(\boldsymbol{p}, t)$ captures the *total population variability*. As shown in Fig. 1(c), the distributional parameters are parameterized by a coordinate-based neural network $f(\cdot)$.

**Architecture.** Both $\mu(\cdot)$ and $\Sigma(\cdot)$ are modeled with a covariate-driven term and a covariate-independent residual:

$$\mu(\boldsymbol{p}, t) = \big[f_\mu(\boldsymbol{p}, t) - f_\mu(\boldsymbol{p}, 0)\big] + h_\mu(\boldsymbol{p}), \quad (3)$$

$$\boldsymbol{L}(\boldsymbol{p}, t) = \big[f_L(\boldsymbol{p}, t) - f_L(\boldsymbol{p}, 0)\big] + h_L(\boldsymbol{p}), \quad (4)$$

where $\boldsymbol{L}$ is the Cholesky factor of $\Sigma$ (see *Covariance Parameterization* below). The subtraction $f(\boldsymbol{p}, t) - f(\boldsymbol{p}, 0)$ ensures that the covariate-driven term vanishes at $t = 0$, so that $\mu(\boldsymbol{p}, 0) = h_\mu(\boldsymbol{p})$ and $\boldsymbol{L}(\boldsymbol{p}, 0) = h_L(\boldsymbol{p})$ by construction; this enforces identifiability between covariate-driven and covariate-independent components.

**Covariance Parameterization.** To ensure positive definiteness, $\Sigma(\boldsymbol{p}, t)$ is parameterized via its Cholesky factor $\boldsymbol{L}$, with $\Sigma = \boldsymbol{L}\boldsymbol{L}^\top$. For shapes in $\mathbb{R}^n$, $\boldsymbol{L}$ has $\frac{n(n+1)}{2}$ free parameters per query point: 3 parameters for 2D geometry and 6 parameters for 3D geometry. A softplus activation on the diagonal entries, followed by adding a small positive constant $\epsilon$, enforces positivity, guaranteeing that $\Sigma$ is symmetric positive definite by construction and enabling stable NLL optimization without explicit constraints.

**Training Objective.** Given a shape $\mathcal{Y}$ represented by $M$ template-displacement pairs $\{(\boldsymbol{p}_j, \boldsymbol{d}_j)\}_{j=1}^M$ associated with a common timestamp $t$, the standard Gaussian negative log-likelihood is

$$\mathcal{L}_{\text{NLL}} = \frac{1}{2M} \sum_{j=1}^M \Big[(\boldsymbol{d}_j - \mu_j)^\top \Sigma_j^{-1} (\boldsymbol{d}_j - \mu_j) + \log \det \Sigma_j\Big], \quad (5)$$

where $\mu_j = \mu(\boldsymbol{p}_j, t)$ and $\Sigma_j = \Sigma(\boldsymbol{p}_j, t)$. Jointly optimizing $\mu$ and $\Sigma$ through Eq. (5) is known to bias the mean estimator, as variance-weighted gradients pull $\mu$ away from the population mean (Skafte et al., 2019; Seitzer et al., 2022). We therefore isolate gradient propagation by training each branch with its own loss:

$$\underbrace{\mathcal{L}_\mu = \tfrac{1}{M} \sum_j \|\boldsymbol{d}_j - \mu_j\|_1}_{\text{trains } \mu}, \quad \underbrace{\mathcal{L}_\Sigma = \mathcal{L}_{\text{NLL}}}_{\text{trains } \Sigma}, \quad (6)$$

with $\mu$ held fixed when computing $\mathcal{L}_\Sigma$. We use $\ell_1$ on $\mu(\cdot)$ for robustness against outliers (Huber, 1992); the underlying model in Eq. (2) remains a Gaussian field.

**Training Strategy.** To ensure stable convergence, we employ a two-stage curriculum. In Stage 1 (epochs 1 to $T_{\text{warm}}$), we freeze the covariance head and train only $\mu$ with $\mathcal{L}_\mu$. In Stage 2, both branches are trained jointly using $\mathcal{L}_\mu + \mathcal{L}_\Sigma$ as defined above. We set $T_{\text{warm}} = 10$.

## 4.2. Individual Intrinsic Time $\hat{\tau}$

We formulate the estimation of individual intrinsic time $\hat{\tau}$ as a Maximum Likelihood Estimation (MLE) problem. Given a template point $\boldsymbol{p}$ and its observed displacement $\boldsymbol{d}$, we seek the best intrinsic time $\hat{\tau}_{\text{MLE}}$ that maximizes the conditional likelihood:

$$\hat{\tau}_{\text{MLE}} = \arg\max_\tau \log p(\boldsymbol{d} \mid \boldsymbol{p}, \tau) \quad (7)$$

where $p(\boldsymbol{d} \mid \boldsymbol{p}, \tau) = \mathcal{N}(\mu(\boldsymbol{p}, \tau), \Sigma(\boldsymbol{p}, \tau))$ is the learned conditional shape distribution as introduced in Sec. 4.1.

Since solving this optimization iteratively for every subject is computationally prohibitive, we employ *amortized inference*. We train an inverse encoder $g(\cdot)$ to directly predict intrinsic time from local observations.

The inverse encoder $g(\cdot)$ is trained using synthetic data generated by the learned forward model $f(\cdot)$. Specifically, we sample template coordinates $\boldsymbol{p}$ and intrinsic times $\tau$ uniformly, and query the forward model $f(\cdot)$ to obtain the corresponding mean displacement $\boldsymbol{d} = \mu(\boldsymbol{p}, \tau)$. This yields training triplets $\{(\boldsymbol{p}, \boldsymbol{d}, \tau)\}$, enabling supervised training of the inverse mapping. The inverse encoder $g(\cdot)$ is then trained with an $L_1$ loss as

$$\mathcal{L}_{\text{inv}} = \frac{1}{M} \sum_{j=1}^M |g(\boldsymbol{p}_j, \boldsymbol{d}_j) - \tau_j|, \quad (8)$$

where $M$ is the number of sampled pairs.

At test time, given a novel subject with observed displacements, the encoder directly maps each template-displacement pair $(\boldsymbol{p}, \boldsymbol{d})$ to an intrinsic time estimate $\hat{\tau} = g(\boldsymbol{p}, \boldsymbol{d})$, enabling dense temporal inference across the anatomy without iterative optimization.

## 4.3. Intrinsic Time Distribution $p(\tau \mid \boldsymbol{p}, t)$

Having established how to estimate individual intrinsic time $\hat{\tau}$, we now characterize its population-level distribution $p(\tau \mid \boldsymbol{p}, t)$. This distribution addresses a fundamental clinical question: for subjects at the same chronological time $t$, how much do their intrinsic times vary? A narrow distribution indicates that chronological time reliably predicts developmental progression, while a wide distribution suggests substantial individual variation. This information is critical for establishing normative ranges and identifying subjects whose development deviates from the population.

We quantify the spread of $p(\tau \mid \boldsymbol{p}, t)$ through the lens of estimation theory (Cover, 1999). The *score function* measures how sensitively the log-likelihood responds to changes in time, defined as

$$U(\boldsymbol{d}; \boldsymbol{p}, t) := \frac{\partial}{\partial t} \log p(\boldsymbol{d} \mid \boldsymbol{p}, t) \tag{9}$$

A large $|U|$ indicates that the observed displacement $\boldsymbol{d}$ provides strong evidence for distinguishing nearby time points. The **Fisher Information** is defined as the expected squared score as

$$I(\boldsymbol{p}, t) := \mathbb{E}_{\boldsymbol{d} \sim p(\boldsymbol{d} \mid \boldsymbol{p}, t)} \left[ U(\boldsymbol{d}; \boldsymbol{p}, t)^2 \right] \tag{10}$$

which quantifies the average amount of information that an observed displacement carries about the time parameter (Cramér, 1999; Rao et al., 1945).

We assume that the population-average intrinsic time equals the chronological time, i.e., $\mathbb{E}_{p(\boldsymbol{d} \mid \boldsymbol{p}, t)}[\tau] = t$. This reflects a standard modeling assumption in longitudinal medical shape analysis that the population mean trajectory is parameterized by chronological age, while individual subjects may follow advanced or delayed progression around this mean (Durrleman et al., 2013; Hong et al., 2014). Under this assumption, $\tau$ can be viewed as an unbiased estimator of $t$, and the Cramér-Rao inequality yields

$$\text{Var}(\tau \mid \boldsymbol{p}, t) \geq \frac{1}{I(\boldsymbol{p}, t)} \tag{11}$$

Eq. (11) reveals a fundamental trade-off: high Fisher Information (shapes that change distinctly over time) enables precise estimation, while low Fisher Information (ambiguous shapes) leads to irreducible uncertainty.

For our heteroscedastic Gaussian model $p(\boldsymbol{d} \mid \boldsymbol{p}, t) = \mathcal{N}(\mu(\boldsymbol{p}, t), \Sigma(\boldsymbol{p}, t))$, the Fisher Information admits a closed-form expression (Skovgaard, 1984; Nielsen, 2023) that decomposes into two terms,

$$I_{\text{full}}(\boldsymbol{p}, t) = \underbrace{\left(\frac{\partial \mu}{\partial t}\right)^\top \Sigma^{-1} \left(\frac{\partial \mu}{\partial t}\right)}_{I_\mu} + \underbrace{\frac{1}{2} \text{tr}\left(\left(\Sigma^{-1} \frac{\partial \Sigma}{\partial t}\right)^2\right)}_{I_\Sigma}. \tag{12}$$

The two terms in Eq. (12) represent two independent sources of information about $t$. $I_\mu$ captures information from the evolution of the mean trajectory, while $I_\Sigma$ captures information from changes in population dispersion. These two sources are statistically independent, following from a classical result in information geometry: the mean and covariance parameters are orthogonal under the Fisher-Rao metric (Amari, 2016; Skovgaard, 1984). Although both are statistically valid, they answer different questions. $I_\mu$ tells us how precisely we can localize an individual along the developmental trajectory, which is our definition of temporal variability. $I_\Sigma$ captures changes in *structural variability* (see Sec. 3), i.e., how anatomical diversity evolves over time; while statistically informative, this is orthogonal to our goal of localizing individuals along the mean developmental trajectory. We therefore retain only $I_\mu$ as Fisher Information

$$I(\boldsymbol{p}, t) := \left(\frac{\partial \mu}{\partial t}\right)^\top \Sigma^{-1} \left(\frac{\partial \mu}{\partial t}\right) \tag{13}$$

The temporal uncertainty is then quantified as

$$\sigma_\tau^2(\boldsymbol{p}, t) \approx I^{-1}(\boldsymbol{p}, t) = \frac{1}{\left(\frac{\partial \mu}{\partial t}\right)^\top \Sigma^{-1} \left(\frac{\partial \mu}{\partial t}\right)} \tag{14}$$

This estimate is analytic: $\frac{\partial \mu}{\partial t}$ comes from automatic differentiation and $\Sigma^{-1}$ from a single forward pass, so unlike Monte Carlo propagation, it incurs no sampling variance and can be queried densely across the anatomy in one pass.

Complete derivations of both the Cramér-Rao in Eq. (11) bound and the closed-form Fisher Information in Eq. (12) are provided in Appendix A.2.

## 4.4. Applications: A Unified Framework for Shape Analysis

**Continuous Shape Trajectory** For any template coordinate $\boldsymbol{p} \in \Omega$ at time $t$, the normative population distribution $\boldsymbol{d} \sim \mathcal{N}(\mu(\boldsymbol{p}, t), \Sigma(\boldsymbol{p}, t))$ provides a reference standard for healthy development at arbitrary spatiotemporal resolutions.

**Intrinsic Time Estimation.** Given an observed anatomy $\mathcal{Y}$, the inverse encoder $g$ (Sec. 4.2) estimates a local intrinsic time $\hat{\tau} = g(\boldsymbol{p}, \boldsymbol{d})$ for each template point $\boldsymbol{p}$, generating a dense map of intrinsic time across the anatomy. To estimate global developmental progression $\bar{\tau}$, one can aggregate these point-level estimates. When the chronological age $t$ is unknown, we simply compute the arithmetic mean:

$$\bar{\tau} = \frac{1}{|\mathcal{T}|} \sum_{\boldsymbol{p} \in \mathcal{T}} g(\boldsymbol{p}, \boldsymbol{d}). \tag{15}$$

When $t$ is available, we can leverage the Fisher Information $I(\boldsymbol{p}, t)$ to obtain a more refined estimate by weighting each contribution according to its temporal discriminability:

$$\bar{\tau} = \frac{\sum_{\boldsymbol{p} \in \mathcal{T}} I(\boldsymbol{p}, t) \cdot g(\boldsymbol{p}, \boldsymbol{d})}{\sum_{\boldsymbol{p} \in \mathcal{T}} I(\boldsymbol{p}, t)}, \tag{16}$$

thereby prioritizing regions with high temporal discriminability while down-weighting ambiguous areas.

**Personalized Longitudinal Prediction**  Let $y_0$ denote the observed position of template point $p$ on anatomy $\mathcal{Y}_0$ at chronological time $t_0$, with estimated intrinsic time $\tau_0$. To forecast the future position $y_1$ at chronological time $t_1 = t_0 + \Delta t$, we assume the subject's temporal z-score remains constant. Specifically, the temporal z-score at $t_0$ is defined as

$$z_\tau = \frac{\tau_0 - t_0}{\sigma_\tau(t_0)}, \qquad (17)$$

where $\sigma_\tau(t_0)$ denotes the population-level temporal variability at $t_0$. Assuming $z_\tau$ remains unchanged, the predicted intrinsic time at $t_1$ is given by

$$\tau_1 = t_1 + z_\tau \cdot \sigma_\tau(t_1). \qquad (18)$$

The future anatomical position is then obtained from the mean trajectory as

$$y_1 = p + \mu(p, \tau_1). \qquad (19)$$

This formulation preserves the subject's developmental progression stage (whether advanced or delayed) while predicting their personalized trajectory.

**OOD Detection.**  We demonstrate OOD detection on pediatric airway obstruction (e.g., subglottic stenosis), where pathological constrictions manifest as regions that appear *developmentally younger* than the rest of the same anatomy. For each template point $p$ with observed displacement $d$, the inverse encoder yields a local intrinsic time $\hat{\tau}_p = g(p, d)$. Let $\tilde{\tau} = \mathrm{median}_{q \in \mathcal{T}} \, \hat{\tau}_q$ be the anatomy-wide median and $p^* = \arg\max_{q \in \mathcal{T}} \hat{\tau}_q$ the most developmentally advanced point. We score each anatomy by the largest temporal lag of any point relative to $p^*$, normalized by local uncertainty:

$$\mathrm{Score_{OOD}} = \min_{p \in \mathcal{T}} \left[ \frac{\hat{\tau}_p - \tilde{\tau}}{\sigma_\tau(p, t)} - \frac{\hat{\tau}_{p^*} - \tilde{\tau}}{\sigma_\tau(p^*, t)} \right], \qquad (20)$$

where $\sigma_\tau(p, t)$ is the local temporal variability from the Fisher Information. Each bracketed term is a z-score-like deviation of a point's intrinsic time from the anatomy median; the subtraction contrasts each point against $p^*$. A strongly negative score flags regions lagging behind the rest of the anatomy beyond normal variation, highlighting pathology without labeled anomaly data.

## 5. Experiments

Our experiments aim to ① validate PRISM's representations and ② demonstrate its clinical utility. On synthetic data with known ground truth, we first verify that PRISM accurately recovers mean shape evolution, estimates local and global intrinsic time, and quantifies temporal uncertainty in shape space. We then apply it to two clinical tasks: personalized longitudinal prediction of developmental trajectories and OOD detection of pathological airways.

### 5.1. Experimental Setup

5.1.1. DATASETS.

We evaluate `PRISM` on four datasets with increasing complexity. More details are available in Appendix B.

**Starman (G&L).**  We generate two variants with known ground-truth intrinsic time: **Starman (G)** with global temporal uncertainty, and **Starman (L)** with spatially-varying temporal uncertainty where arms and legs follow distinct developmental trajectories. Each variant contains 1,000 training and 1,000 testing subjects with 1–9 longitudinal observations per subject (∼5,000 shapes total per variant). The time $t$ represents a unitless simulation time ranging from 0 to 1.

**ANNY.**  We use ANNY (Brégier et al., 2025), a parametric human body model spanning infancy to adulthood. We simulate inter-subject variability with temporal uncertainty increasing during puberty, following bone age literature (Cole et al., 2010; Thodberg et al., 2008). The dataset contains 1,507 training shapes from 1,000 subjects and 324 testing shapes from 100 subjects. The time $t$ is chronological age in years (0–20).

**Pediatric Airway.**  Our clinical dataset comprises 358 CT scans from 264 subjects (ages 0–19 years), including 34 with longitudinal sequences. Airway surfaces are segmented from CT images using a two-stage UNet (Çiçek et al., 2016). The airways are straightened via a rotation-minimizing frame (Bishop, 1975) to remove pose variation while preserving cross-sectional geometry. An additional 31 scans with subglottic stenosis (pathological airway narrowing) serve as out-of-distribution (OOD) samples for anomaly detection. The time $t$ is chronological age in months; unlike synthetic data, ground-truth intrinsic time is unavailable.

For these three datasets, point samples and SDF values are computed following (Park et al., 2019; Sitzmann et al., 2020). During preprocessing, our template-based registration module learns a shared template and an invertible deformation mapping the template to each anatomy (Appendix A.1); the resulting deformations serve as the training signal for `PRISM`.

5.1.2. COMPARISON METHODS.

We compare with implicit representation methods capable of covariate-conditioned shape modeling. We select A-SDF (Mu et al., 2021), which directly models the signed distance field conditioned on covariates, and NAISR (Jiao et al., 2024), which learns neural deformations from a shared template conditioned on covariates. Both methods use per-shape latent codes to capture individual variation but do not model uncertainty. A detailed architectural and modeling

*Table 1.* **Quantitative evaluation of mean trajectory reconstruction.** CD = Chamfer distance. EMD = Earth mover's distance. HD = Hausdorff distance. Given physical time $t$, each method reconstructs the mean shape $\boldsymbol{\mu}(p, t)$. Metrics are scaled by 100; lower is better. Best results are in **bold**.

| Dataset | Method | CD ($\downarrow$) | HD ($\downarrow$) | EMD ($\downarrow$) |
|---------|--------|--------|--------|--------|
| | A-SDF | **0.137** | **6.842** | 0.215 |
| Starman (G) | NAISR | 0.140 | 6.862 | 0.216 |
| | PRISM | 0.138 | 6.907 | **0.195** |
| | A-SDF | 0.288 | **9.408** | 0.374 |
| Starman (L) | NAISR | 0.351 | 10.128 | 0.401 |
| | PRISM | **0.287** | 9.481 | **0.360** |
| | A-SDF | **0.040** | 4.586 | **0.997** |
| ANNY | NAISR | 0.058 | 8.158 | 1.165 |
| | PRISM | 0.048 | **4.450** | 1.124 |
| | A-SDF | 0.114 | 10.508 | 2.040 |
| Airway | NAISR | 0.072 | 10.075 | 1.422 |
| | PRISM | **0.064** | **9.614** | **1.308** |

comparison of PRISM with NAISR and A-SDF is provided in Appendix B.6.

## 5.2. Experiment Results

**Population Trend.** As summarized in Tab. 1, PRISM consistently achieves competitive or superior performance across all four datasets. On the Airway dataset, our method attains the lowest reconstruction error while remaining comparable to the best on Starman (G&L) and ANNY. Compared against NAISR specifically, PRISM improves on nearly all metrics across all datasets. We attribute this gain to its use of precomputed dense correspondences: by decoupling correspondence from reconstruction, PRISM optimizes a simpler objective, whereas NAISR must learn both jointly, which can complicate optimization. A-SDF, in contrast, directly maps covariates to signed distance values without explicit correspondence. This formulation fits the mean trajectory well given sufficient data (Starman (G&L), ANNY), but overfits on the smaller dataset (Airway), yielding higher reconstruction errors than deformation-based methods (Jiao et al., 2024).

This quantitative comparison confirms that PRISM provides a high-fidelity estimation of the "average" anatomy, effectively recovering the global developmental trend from sparse, unstructured cross-sectional observations.

Qualitatively, as demonstrated in Fig. 3 and Fig. 8, the estimated mean trajectory exhibits smooth, biologically plausible transitions across the developmental timeline, demonstrating that PRISM successfully interpolates the underlying population-level growth dynamics.

**Intrinsic Time Estimation.** We evaluate PRISM's intrinsic time estimation at both global and local levels. The inverse encoder $g(\cdot)$ estimates intrinsic time in a single for-

*Table 2.* **Global intrinsic time estimation.** Pearson correlation ($r \uparrow$), $R^2$ ($\uparrow$), mean absolute error (MAE $\downarrow$), and per-case inference time. PRISM (Amortized) remains competitive with baselines while being an order of magnitude faster.

| Dataset | Method | Inference | $r$ ($\uparrow$) | $R^2$ ($\uparrow$) | MAE ($\downarrow$) | Time/case (s) ($\downarrow$) |
|---------|--------|-----------|--------|--------|--------|--------|
| | A-SDF | TTO | 0.992 | 0.983 | 0.016 | 4.005 |
| Starman (G) | NAISR | TTO | 0.991 | 0.980 | 0.019 | 7.892 |
| | PRISM | Amortized | **1.000** | **0.999** | **0.005** | **0.040** |
| | A-SDF | TTO | **0.996** | **0.991** | **0.351**$_{yr.}$ | 7.045 |
| ANNY | NAISR | TTO | 0.988 | 0.958 | 0.933$_{yr.}$ | 8.606 |
| | PRISM | Amortized | 0.993 | 0.971 | 0.602$_{yr.}$ | **0.430** |
| | A-SDF | TTO | 0.745 | 0.334 | 42.886$_{mo.}$ | 16.863 |
| Airway | NAISR | TTO | 0.908 | 0.820 | 20.512$_{mo.}$ | 24.363 |
| | PRISM | Amortized | 0.893 | 0.792 | 22.639$_{mo.}$ | **0.805** |
| | PRISM | TTO | **0.923** | **0.843** | **19.618**$_{mo.}$ | 8.838 |

*Table 3.* **Local intrinsic time estimation.** On Starman (L), intrinsic time varies across body parts; PRISM is the only method applicable.

| Dataset | Method | Location | $r$ ($\uparrow$) | $R^2$ ($\uparrow$) | MAE ($\downarrow$) |
|---------|--------|----------|--------|--------|--------|
| Starman (L) | PRISM | Arm | 1.000 | 0.999 | 0.008 |
| | | Leg | 1.000 | 0.999 | 0.004 |

ward pass (Eq. (15)). In contrast, baselines A-SDF and NAISR require per-sample test-time optimization, denoted TTO (Jiao et al., 2024), which treats time as a free variable while keeping network weights frozen. On synthetic datasets, ground-truth intrinsic time is available; on the Airway dataset, chronological age serves as a surrogate.

*Global estimation.* The amortized PRISM matches or exceeds baseline accuracy while being an order of magnitude faster (Tab. 2). On Airway, NAISR with TTO edges out amortized PRISM, but this gap reflects the amortization trade-off rather than a methodological disadvantage.

*Local estimation.* On Starman (L), intrinsic time varies spatially across body parts, a regime where baselines are inapplicable since they only estimate a single global time per shape. As shown in Tab. 3, PRISM achieves near-perfect estimation for both arm and leg regions, demonstrating its unique capability to capture fine-grained developmental heterogeneity.

**Temporal Uncertainty.** We quantify temporal uncertainty via the Fisher information (Eq. (14)). We first validate on the synthetic Starman (G&L) datasets, where the ground truth is available. As shown in Fig. 2, PRISM accurately recovers the true conditional distribution $p(\tau \mid \boldsymbol{p}, t)$ under both global uncertainty (Starman (G)) and local uncertainty (Starman (L)). The predicted 95% confidence intervals (red) align tightly with the ground truth (blue). Minor discrepancies near the temporal boundaries are likely due to fewer training samples at the extremes.

We extend this analysis to the airway dataset in Fig. 4. Since no ground truth temporal uncertainty is available, we overlay

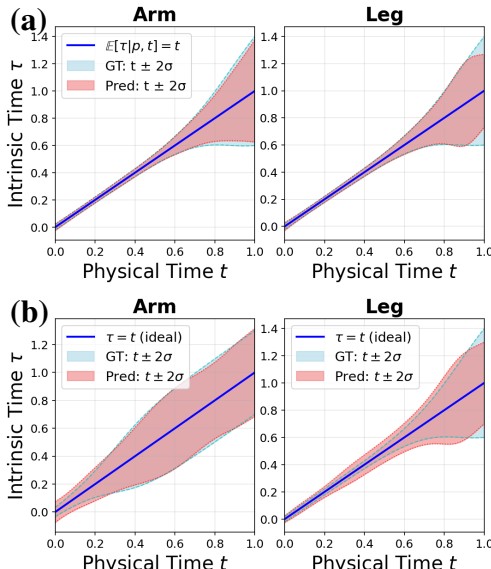

**Figure 2.** Qualitative validation of uncertainty estimation on the simulated Starman datasets. **(a)** Results on the Starman (G) dataset and **(b)** results on the Starman (L) dataset. The plot compares the uncertainty estimates from PRISM at different locations with the ground truth. The blue shaded regions represent the ground truth distribution of the conditional distribution $p(\tau \mid \boldsymbol{p}, t)$, while the red shaded regions show the distribution estimated by our method. The tight alignment between the two demonstrates PRISM's ability to accurately recover the true underlying uncertainty profile of the data-generating process.

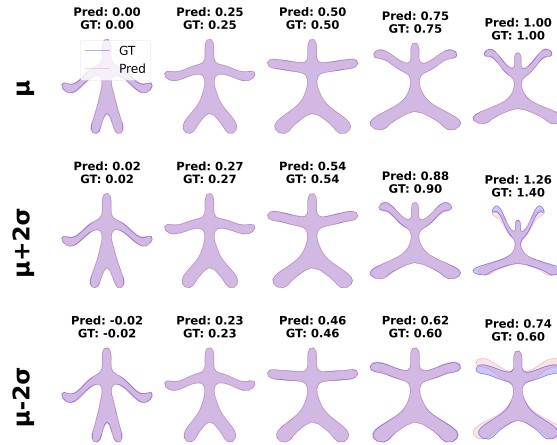

**Figure 3.** Visual comparison of uncertainty-aware shape reconstruction on the Starman (G) dataset. We decode shapes using time points at the mean and $\pm 2\sigma$ of the predicted intrinsic time distribution. The red contours represent the shapes reconstructed by PRISM at these time points, while the blue contours represent the ground truth shapes. The high degree of overlap verifies that the learned temporal uncertainty correctly translates into valid geometrical deformations.

*Table 4.* **Evaluation of personalized shape prediction.**

| Dataset | Method | CD ($\downarrow$) | HD ($\downarrow$) | EMD ($\downarrow$) |
|---|---|---|---|---|
| | A-SDF | 0.165 | 6.818 | 0.189 |
| Starman (G) | NAISR | 0.130 | 6.809 | 0.185 |
| | PRISM | **0.072** | **6.273** | **0.092** |
| | A-SDF | 0.467 | 8.742 | 0.351 |
| Starman (L) | NAISR | 0.595 | 11.143 | 0.450 |
| | PRISM | **0.099** | **8.178** | **0.148** |
| | A-SDF | 0.081 | 4.170 | 0.866 |
| ANNY | NAISR | 0.044 | 7.657 | 1.042 |
| | PRISM | **0.018** | **3.017** | **0.560** |
| | A-SDF | 0.123 | 10.762 | 2.034 |
| Airway | NAISR | 0.065 | 9.599 | 1.351 |
| | PRISM | **0.059** | **9.189** | **1.243** |

actual data points, with chronological age $t$ on the x-axis and predicted intrinsic time $\hat{\tau}$ on the y-axis. Most normal cases lie within the predicted intervals, validating the calibration of our uncertainty estimates. The uncertainty bands vary substantially across anatomical locations, demonstrating PRISM's ability to capture spatially heterogeneous uncertainty. Soft tissue regions such as the base of the tongue exhibit wider bands, consistent with Zhou et al. (2021), who reported high intra-individual variation in these regions due to anatomical variability and imaging sensitivity.

To verify that the learned temporal uncertainty produces meaningful geometric variations, we visualize reconstructed shapes at the bounds of the predicted distribution. As shown in Fig. 3, we decode shapes at the mean ($\mu$) and at $\mu \pm 2\sigma$ (i.e., two standard deviations from the mean). The predicted contours (red) align well with the ground truth (blue), confirming that both the learned mean trajectory and uncertainty estimates are accurate.

As shown in Fig. 8, we decode shapes at $\mu$ and $\mu \pm 2\sigma$ and report the corresponding airway volumes. Almost all subjects with complete airways fall within this range, validating that both the mean trajectory and temporal uncertainty learned by PRISM are reasonable for clinical data.

**Personalized Shape Prediction.** Given a shape at time $t$, we predict its future evolution by estimating the intrinsic time $\hat{\tau}$ and propagating it forward along the learned trajectory following Eq. (19). For the baselines, a global $\hat{\tau}$ is optimized at test time, whereas PRISM leverages pointwise estimates directly from the inverse encoder. As shown in Tab. 4, PRISM performs best on longitudinal prediction. This task requires extrapolating each subject forward in time, where A-SDF—which maps covariates to signed-distance values without a geometric prior—degrades sharply beyond its training range, whereas template-based deformable representations (NAISR, PRISM) remain bounded.

**OOD Detection.** As shown in Tab. 5, PRISM with local scoring substantially outperforms all global-time methods. Notably, PRISM (Global) underperforms NAISR, yet

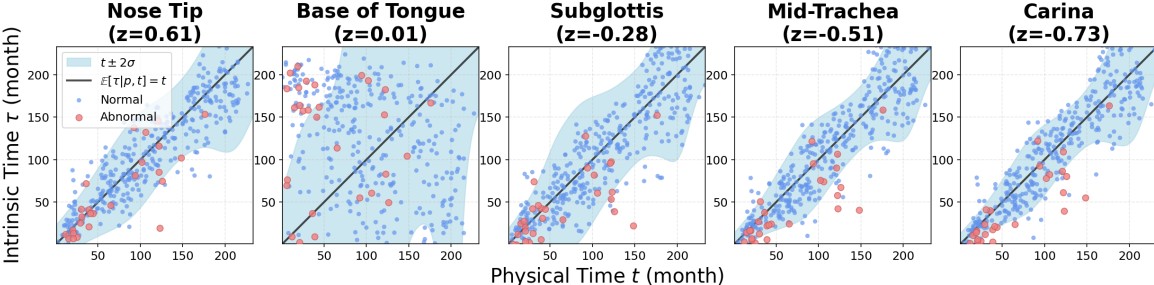

*Figure 4.* Spatially-varying uncertainty quantification across anatomical landmarks in pediatric airways. Each subplot displays the relationship between chronological age $t$ (x-axis) and the predicted intrinsic developmental age $\hat{\tau} = g(\boldsymbol{p}, \boldsymbol{d})$ (y-axis) at a specific anatomical landmark, where $g(\cdot)$ is the learned inverse encoder. Landmarks progress from the nose tip (top-left) to the carina (bottom-right). The shaded region indicates $\tau \pm 2\sigma$ estimated by PRISM; the diagonal line represents $\mathbb{E}[\tau|t] = t$. Blue points denote normal subjects; red points denote abnormal cases.

*Table 5.* **Evaluation of OOD detection on Pediatric Airway dataset.**

| Method | Time $\tau$ | AUC (↑) | Sens (↑) | Spec (↑) | Acc (↑) |
|--------|-------------|---------|----------|----------|---------|
| A-SDF | Global | 0.270 | **1.000** | 0.000 | 0.185 |
| NAISR | Global | 0.605 | 0.613 | 0.650 | 0.643 |
| PRISM | Global | 0.502 | 0.774 | 0.401 | 0.470 |
| PRISM | Local | **0.875** | 0.806 | **0.869** | **0.857** |

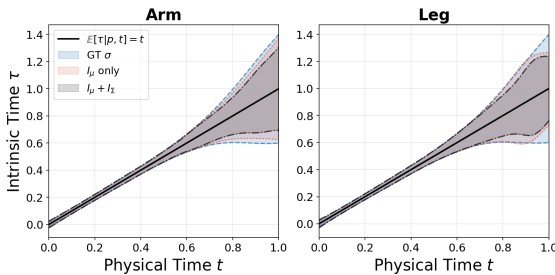

*Figure 5.* **Ablation of $I_\mu$ vs. $I_{\text{full}}$ on Starman (G).** Uncertainty bands from $I_\mu$ (red) and $I_{\text{full}} = I_\mu + I_\Sigma$ (gray), compared to ground truth (blue) at the arm (left) and leg (right) control points. $I_\mu$ tracks the ground truth, while $I_{\text{full}}$ underestimates variability: $I_\Sigma$ tightens the bound but does not capture our target—localizing individuals along the mean trajectory.

PRISM (Local) achieves the best results. This highlights a limitation of global approaches: airway development varies across individuals, anatomical locations, and time, which a single global rate cannot capture. Local estimates detect deviations within each subject's own anatomy, avoiding inter-individual confounds. More details about baseline adaptions for OOD is available in Appendix B.4.

**Empirical Validation: $I_\mu$ vs $I_{\text{full}}$.** We empirically verify our choice of $I_\mu$ over the full Fisher Information $I_{\text{full}} = I_\mu + I_\Sigma$ as the temporal uncertainty metric. On Starman (G), where ground-truth uncertainty is available, the $I_\mu$-derived bands closely track the ground truth, while $I_{\text{full}}$ systematically underestimates the true variability (Fig. 5). This

confirms our choice: $I_\Sigma$ tightens the Cramér-Rao bound by accounting for changes in population dispersion, but this does not reflect our target quantity—the precision of localizing an individual along the mean trajectory. We therefore use $I_\mu$ alone as the temporal uncertainty metric.

## 6. Limitations and Future Work

First, our framework currently conditions shape evolution on a single scalar covariate (e.g., chronological age). Extending our information-theoretic approach to high-dimensional covariate spaces is essential future work. Second, while our experiments demonstrate robust anomaly detection in pediatric airways, the model's capacity for long-term forecasting in degenerative conditions remains underexplored. Stochasticity often increases with biological age, making trajectory prediction increasingly difficult. Future work will focus on personalized predictive modeling for aging-related diseases, leveraging PRISM's heteroscedastic uncertainty maps to predict subject-specific disease progression.

## 7. Conclusion

We introduced PRISM, the first implicit neural representation framework for uncertainty-aware statistical shape analysis. Our approach captures conditional shape distributions, estimates intrinsic developmental time, and quantifies spatially varying temporal uncertainty. A key theoretical contribution is the closed-form Fisher Information metric for analytically quantifying temporal uncertainty, which can be efficiently computed via automatic differentiation inherent to implicit neural representations. Experiments on both synthetic and clinical datasets demonstrate that PRISM is a versatile framework that delivers consistent performance across diverse medical tasks, including developmental forecasting and anomaly detection.

## Acknowledgments

This research was, in part, funded by the National Institutes of Health (NIH) under 1R21HL172230-01A1. The views and conclusions contained in this document are those of the authors and should not be interpreted as representing official policies, either expressed or implied, of the NIH.

## Impact Statement

PRISM was motivated by critical challenges in pediatric airway analysis: modeling longitudinal shape change, quantifying population-level uncertainty conditioned on covariates such as age, and detecting out-of-distribution cases. However, the method extends well beyond airways, applying to other anatomical shape analysis problems. More generally, whenever a covariate's effect on geometry is uncertain and the geometry is represented by an implicit neural representation (INR), PRISM's Fisher-information-based formulation provides a principled framework to propagate uncertainty from outputs back to inputs.

While our current validation focuses on growth-driven shape change—where the covariate exerts a dominant effect on geometry—PRISM's predictions should be interpreted with caution when individual variation approaches or exceeds covariate-driven variation. Disentangling individual-specific traits from covariate-driven dynamics remains future work. Consequently, clinical translation must adhere to these validation boundaries and requires external validation on the target population.

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

# A. Theory and Methods

*Table 6.* Summary of notation.

| Symbol | Description |
|---|---|
| *Data & Geometry* | |
| $N$ | Number of shape observations |
| $\mathcal{Y}_i$ | The $i$-th observed shape |
| $\mathcal{T}$ | Reference template shape |
| $\Omega \subset \mathbb{R}^3$ | Canonical domain |
| $\boldsymbol{p}$ | Template coordinate |
| $\boldsymbol{d}$ | Displacement vector |
| $\boldsymbol{y}$ | Target position ($\boldsymbol{y} = \boldsymbol{p} + \boldsymbol{d}$) |
| $\boldsymbol{\phi}_i$ | Displacement field for subject $i$ |
| *Time Variables* | |
| $t$ | Chronological time (observed covariate) |
| $\tau$ | Intrinsic time (latent developmental stage) |
| $\hat{\tau}$ | Estimated intrinsic time |
| *Distribution Parameters* | |
| $\mu(\boldsymbol{p}, t)$ | Mean displacement trajectory |
| $\Sigma(\boldsymbol{p}, t)$ | Covariance matrix |
| $p(\boldsymbol{d} \mid \boldsymbol{p}, t)$ | Conditional displacement distribution |
| *Fisher Information* | |
| $U$ | Score function |
| $I(\boldsymbol{p}, t)$ | Fisher Information |
| $\sigma_\tau^2(\boldsymbol{p}, t)$ | Temporal variability ($\approx I^{-1}$) |
| *Networks* | |
| $f(\cdot)$ | Forward model (outputs $\boldsymbol{\mu}, \boldsymbol{\Sigma}$) |
| $g(\cdot)$ | Inverse encoder (outputs $\hat{\tau}$) |

## A.1. Template-based Point Correspondence

All components of PRISM rely on dense point correspondence across subjects, where each template coordinate $\boldsymbol{p}$ maps to anatomically equivalent locations across all shapes. We establish this correspondence via a template-based registration module that learns invertible deformations between a shared template and individual anatomies. After training, the learned deformations serve as pseudo ground-truth supervision for PRISM, which predicts displacements conditioned on covariates such as age.

**Architecture.** The registration model employs two hypernetwork-based architectures to model bidirectional deformations. For each training instance $\mathcal{Y}_i$, we learn two latent codes: $\boldsymbol{z}_i^{\mathrm{fwd}} \in \mathbb{R}^{1024}$ for the forward (target-to-template) direction and $\boldsymbol{z}_i^{\mathrm{bwd}} \in \mathbb{R}^{1024}$ for the backward (template-to-target) direction. Each hypernetwork $\mathcal{H}$ takes a latent code as input and outputs the weights of a SIREN-based displacement network $\mathcal{D}$ (Deng et al., 2021). Unlike DIF-Net, we learn two separate networks to ensure invertibility.

For the forward direction, given a point $\boldsymbol{y}$ in the target space, the corresponding point in template space $\boldsymbol{p}$ is computed as

$$\boldsymbol{p} = \Phi_{\mathrm{fwd}}(\boldsymbol{y}) = \boldsymbol{y} + \mathcal{D}_{\mathrm{fwd}}(\boldsymbol{y}; \mathcal{H}_{\mathrm{fwd}}(\boldsymbol{z}_i^{\mathrm{fwd}})). \tag{21}$$

For the backward direction, given a point $\boldsymbol{p}$ in the template space, the recovered target point $\boldsymbol{y}'$ is computed as

$$\boldsymbol{y}' = \Phi_{\mathrm{bwd}}(\boldsymbol{p}) = \boldsymbol{p} + \mathcal{D}_{\mathrm{bwd}}(\boldsymbol{p}; \mathcal{H}_{\mathrm{bwd}}(\boldsymbol{z}_i^{\mathrm{bwd}})). \tag{22}$$

A shared template network $\mathcal{T}(\cdot)$ maps points in the template space to signed distance values, implicitly representing the canonical anatomy. The composed mapping $\mathcal{T} \circ \Phi_{\mathrm{fwd}} : \boldsymbol{y} \mapsto \mathcal{T}(\Phi_{\mathrm{fwd}}(\boldsymbol{y}))$ thus predicts the signed distance for any query point $\boldsymbol{y}$ in the target space.

**Training Objectives.** The registration model is trained with SDF reconstruction losses and invertibility constraints, following the formulation in SIREN (Sitzmann et al., 2020).

*SDF Reconstruction Loss.* For each training sample, we are given a set of query points $\{\boldsymbol{y}_j\}$ sampled in the target space, along with their ground-truth signed distance values $\{s_j\}$ and surface normals $\{\boldsymbol{n}_j\}$. The composed mapping $\mathcal{T} \circ \Phi_{\text{fwd}}$ first deforms each query point $\boldsymbol{y}_j$ to the template space, then evaluates the template SDF, yielding the predicted signed distance $\hat{s}_j = (\mathcal{T} \circ \Phi_{\text{fwd}})(\boldsymbol{y}_j)$. The SDF reconstruction loss minimizes the discrepancy between the predicted and ground-truth signed distances as

$$\mathcal{L}_{\text{sdf}} = \frac{1}{M} \sum_j |(\mathcal{T} \circ \Phi_{\text{fwd}})(\boldsymbol{y}_j) - s_j|. \tag{23}$$

*Normal Alignment Loss.* The gradient of the composed mapping $\mathcal{T} \circ \Phi_{\text{fwd}}$ with respect to the query point $\boldsymbol{y}_j$ should align with the ground-truth surface normal $\boldsymbol{n}_j$. The normal alignment loss penalizes deviations from perfect alignment as

$$\mathcal{L}_{\text{norm}} = \frac{1}{M} \sum_j \left( 1 - \frac{\nabla_{\boldsymbol{y}}(\mathcal{T} \circ \Phi_{\text{fwd}})(\boldsymbol{y}_j) \cdot \boldsymbol{n}_j}{\|\nabla_{\boldsymbol{y}}(\mathcal{T} \circ \Phi_{\text{fwd}})(\boldsymbol{y}_j)\| \, \|\boldsymbol{n}_j\|} \right). \tag{24}$$

*Eikonal Regularization.* A valid signed distance function must satisfy unit gradient norm. The eikonal loss enforces this constraint as

$$\mathcal{L}_{\text{eik}} = \frac{1}{M} \sum_j \big|\|\nabla_{\boldsymbol{y}}(\mathcal{T} \circ \Phi_{\text{fwd}})(\boldsymbol{y}_j)\| - 1\big|. \tag{25}$$

*Invertibility Constraints.* The above SDF losses only constrain the forward deformation $\Phi_{\text{fwd}}$. To ensure the backward deformation $\Phi_{\text{bwd}}$ is the true inverse, we employ cycle consistency losses. Let $\Phi_{\text{rt}} = \Phi_{\text{bwd}} \circ \Phi_{\text{fwd}}$ denote the round-trip transformation that maps a target point $\boldsymbol{y}$ to template space and back. For a perfect inverse, $\Phi_{\text{rt}}(\boldsymbol{y}) = \boldsymbol{y}$.

The ICON loss (Greer et al., 2021) directly penalizes the round-trip displacement as

$$\mathcal{L}_{\text{icon}} = \frac{1}{M} \sum_j \|\Phi_{\text{rt}}(\boldsymbol{y}_j) - \boldsymbol{y}_j\|_2. \tag{26}$$

The GradICON loss (Tian et al., 2023) enforces that the Jacobian of the round-trip transformation equals identity as

$$\mathcal{L}_{\text{grad}} = \frac{1}{M} \sum_j \|\nabla_{\boldsymbol{y}} \Phi_{\text{rt}}(\boldsymbol{y}_j) - \boldsymbol{I}\|_F. \tag{27}$$

Both losses are necessary: GradICON alone ensures locally correct derivatives but can admit a global translation offset, while ICON anchors the absolute positions.

*Total Loss.* The total training objective is

$$\mathcal{L}_{\text{total}} = \lambda_{\text{sdf}}\mathcal{L}_{\text{sdf}} + \lambda_{\text{norm}}\mathcal{L}_{\text{norm}} + \lambda_{\text{eik}}\mathcal{L}_{\text{eik}} + \lambda_{\text{icon}}\mathcal{L}_{\text{icon}} + \lambda_{\text{grad}}\mathcal{L}_{\text{grad}}, \tag{28}$$

with $\lambda_{\text{sdf}} = 3000$, $\lambda_{\text{norm}} = 100$, $\lambda_{\text{eik}} = 50$, and $\lambda_{\text{icon}} = \lambda_{\text{grad}} = 100$.

After training, we extract the template-to-target deformations $\Phi_{\text{bwd}}$ as pseudo ground-truth for training `PRISM`. Specifically, for each subject $\mathcal{Y}_i$ with chronological time $t_i$, we sample template points $\boldsymbol{p}$ and compute the displacement as $\boldsymbol{d} = \Phi_{\text{bwd}}^{(i)}(\boldsymbol{p}) - \boldsymbol{p}$, yielding training pairs $\{(\boldsymbol{p}, \boldsymbol{d}, t_i)\}$ for `PRISM` as shown in Fig. 7(c).

## A.2. Derivation of the Fisher Information Metric

This appendix provides self-contained derivations of the Fisher Information metric and the Cramér-Rao lower bound for the heteroscedastic Gaussian model introduced in Sec. 4. The derivations follow standard techniques from estimation theory (Cramér, 1999; Rao et al., 1945), information geometry (Amari, 2016), and matrix calculus (Petersen et al., 2008). The closed-form Fisher Information for multivariate normal distributions is a classical result (Skovgaard, 1984; Nielsen, 2023).

### A.2.1. SCORE FUNCTION DERIVATION

We begin with the conditional distribution of displacement $\boldsymbol{d}$ at template coordinate $\boldsymbol{p}$ given chronological time $t$:

$$p(\boldsymbol{d} \mid \boldsymbol{p}, t) = \mathcal{N}(\mu(\boldsymbol{p}, t), \Sigma(\boldsymbol{p}, t)). \tag{29}$$

The log-likelihood function for $p(\boldsymbol{d} \mid \boldsymbol{p}, t)$ is

$$\ell(\boldsymbol{d}; \boldsymbol{p}, t) = \log p(\boldsymbol{d} \mid \boldsymbol{p}, t) = -\frac{1}{2}(\boldsymbol{d} - \boldsymbol{\mu})^\top \Sigma^{-1}(\boldsymbol{d} - \boldsymbol{\mu}) - \frac{1}{2}\log \det \Sigma - \frac{3}{2}\log(2\pi), \tag{30}$$

where $\boldsymbol{\mu} := \mu(\boldsymbol{p}, t)$ denotes the mean vector and $\Sigma := \Sigma(\boldsymbol{p}, t)$ denotes the covariance matrix, both depending on $t$. We denote their temporal derivatives as $\boldsymbol{\mu}_t := \partial_t \mu(\boldsymbol{p}, t)$ and $\Sigma_t := \partial_t \Sigma(\boldsymbol{p}, t)$.

To derive the score function $U(\boldsymbol{d}; \boldsymbol{p}, t) := \partial_t \ell(\boldsymbol{d}; \boldsymbol{p}, t)$, we utilize two standard matrix derivative identities:

$$\partial_t \Sigma^{-1} = -\Sigma^{-1}\Sigma_t\Sigma^{-1}, \tag{31}$$

$$\partial_t \log \det \Sigma = \mathrm{tr}(\Sigma^{-1}\Sigma_t). \tag{32}$$

Define the residual $\boldsymbol{r} := r(\boldsymbol{p}, t) = \boldsymbol{d} - \mu(\boldsymbol{p}, t)$. Since $\boldsymbol{d}$ is fixed during differentiation, $\partial_t \boldsymbol{r} = -\boldsymbol{\mu}_t$.

We write the log-likelihood as

$$\ell(\boldsymbol{d}; \boldsymbol{p}, t) = -\frac{1}{2}Q(\boldsymbol{p}, t) - \frac{1}{2}\log \det \Sigma(\boldsymbol{p}, t) + C, \tag{33}$$

where $Q(\boldsymbol{p}, t) := \boldsymbol{r}^\top \Sigma^{-1} \boldsymbol{r}$. To derive the score function $U(\boldsymbol{d}; \boldsymbol{p}, t)$, we differentiate $Q$ using the product rule as

$$\begin{aligned}
\partial_t Q(\boldsymbol{p}, t) &= (\partial_t \boldsymbol{r})^\top \Sigma^{-1} \boldsymbol{r} + \boldsymbol{r}^\top (\partial_t \Sigma^{-1}) \boldsymbol{r} + \boldsymbol{r}^\top \Sigma^{-1}(\partial_t \boldsymbol{r}) \\
&= (-\boldsymbol{\mu}_t)^\top \Sigma^{-1} \boldsymbol{r} + \boldsymbol{r}^\top (-\Sigma^{-1}\Sigma_t\Sigma^{-1}) \boldsymbol{r} + \boldsymbol{r}^\top \Sigma^{-1}(-\boldsymbol{\mu}_t) \\
&= -2\boldsymbol{\mu}_t^\top \Sigma^{-1} \boldsymbol{r} - \boldsymbol{r}^\top \Sigma^{-1}\Sigma_t\Sigma^{-1} \boldsymbol{r},
\end{aligned} \tag{34}$$

where we used the symmetry of the quadratic form. The score function is therefore

$$U(\boldsymbol{d}; \boldsymbol{p}, t) = \boldsymbol{\mu}_t^\top \Sigma^{-1} \boldsymbol{r} + \frac{1}{2}\boldsymbol{r}^\top \Sigma^{-1}\Sigma_t\Sigma^{-1} \boldsymbol{r} - \frac{1}{2}\mathrm{tr}(\Sigma^{-1}\Sigma_t). \tag{35}$$

### A.2.2. FISHER INFORMATION COMPUTATION

The Fisher Information quantifies how much information an observation $\boldsymbol{d}$ carries about the time parameter $t$. At $(\boldsymbol{p}, t)$, it is defined as the expected squared score:

$$I(\boldsymbol{p}, t) := \mathbb{E}_{\boldsymbol{d} \sim p(\boldsymbol{d} \mid \boldsymbol{p}, t)}\left[U(\boldsymbol{d}; \boldsymbol{p}, t)^2\right]. \tag{36}$$

Since $\boldsymbol{r} = \boldsymbol{d} - \boldsymbol{\mu} \sim \mathcal{N}(\boldsymbol{0}, \Sigma)$, we can express the score function (Eq. (35)) in terms of the centered residual $\boldsymbol{r}$. For notational convenience, we define:

$$A := \Sigma^{-1}\Sigma_t\Sigma^{-1}, \tag{37}$$

$$B := \frac{1}{2}\mathrm{tr}(\Sigma^{-1}\Sigma_t). \tag{38}$$

The score function then decomposes as

$$U(\boldsymbol{r}) = U_{\text{linear}}(\boldsymbol{r}) + U_{\text{quadratic}}(\boldsymbol{r}), \tag{39}$$

where

$$U_{\text{linear}}(\boldsymbol{r}) := \boldsymbol{\mu}_t^\top \Sigma^{-1} \boldsymbol{r}, \tag{40}$$

$$U_{\text{quadratic}}(\boldsymbol{r}) := \frac{1}{2}\boldsymbol{r}^\top A \boldsymbol{r} - B. \tag{41}$$

Expanding the square, the Fisher Information becomes

$$I(\boldsymbol{p}, t) = \mathbb{E}[U^2] = \mathbb{E}[U_{\text{linear}}^2] + 2\mathbb{E}[U_{\text{linear}}U_{\text{quadratic}}] + \mathbb{E}[U_{\text{quadratic}}^2]. \tag{42}$$

We evaluate each term separately.

**Cross-Term Vanishes**   We first show that the mixed term $\mathbb{E}[U_{\text{linear}}U_{\text{quadratic}}]$ equals zero. Since $B$ is a constant and $\mathbb{E}[U_{\text{linear}}] = \boldsymbol{\mu}_t^\top \Sigma^{-1} \mathbb{E}[\boldsymbol{r}] = 0$, we have

$$\mathbb{E}[U_{\text{linear}}U_{\text{quadratic}}] = \mathbb{E}\left[U_{\text{linear}} \cdot \left(\frac{1}{2}\boldsymbol{r}^\top A \boldsymbol{r} - B\right)\right] = \frac{1}{2}\mathbb{E}\left[U_{\text{linear}} \cdot \boldsymbol{r}^\top A \boldsymbol{r}\right]. \tag{43}$$

To evaluate this expectation, we write $U_{\text{linear}}$ in component form. Let $\boldsymbol{b} := \Sigma^{-1}\boldsymbol{\mu}_t$, so that

$$U_{\text{linear}} = \boldsymbol{b}^\top \boldsymbol{r} = \sum_{k=1}^{3} b_k r_k. \tag{44}$$

Similarly, the quadratic form expands as

$$\boldsymbol{r}^\top A \boldsymbol{r} = \sum_{i=1}^{3}\sum_{j=1}^{3} A_{ij} r_i r_j. \tag{45}$$

Therefore,

$$\mathbb{E}\left[U_{\text{linear}} \cdot \boldsymbol{r}^\top A \boldsymbol{r}\right] = \sum_{i,j,k} b_k A_{ij} \mathbb{E}[r_k r_i r_j]. \tag{46}$$

For a centered multivariate Gaussian $\boldsymbol{r} \sim \mathcal{N}(\boldsymbol{0}, \Sigma)$, all odd-order central moments vanish. In particular, the third-order moment satisfies

$$\mathbb{E}[r_k r_i r_j] = 0 \quad \text{for all } i, j, k. \tag{47}$$

This follows from the symmetry of the Gaussian distribution: for any odd function $f(\boldsymbol{r})$, we have $\mathbb{E}[f(\boldsymbol{r})] = 0$. Since $U_{\text{linear}}$ is linear (odd) in $\boldsymbol{r}$ and $\boldsymbol{r}^\top A \boldsymbol{r}$ is quadratic (even) in $\boldsymbol{r}$, their product is an odd function, yielding

$$\boxed{\mathbb{E}[U_{\text{linear}}U_{\text{quadratic}}] = 0.} \tag{48}$$

**Linear Term Contribution**   We now compute $\mathbb{E}[U_{\text{linear}}^2]$. Since $\mathbb{E}[\boldsymbol{r}] = \boldsymbol{0}$, we have $\mathbb{E}[U_{\text{linear}}] = 0$, thus

$$\mathbb{E}[U_{\text{linear}}^2] = \text{Var}(U_{\text{linear}}). \tag{49}$$

Recall that $U_{\text{linear}} = \boldsymbol{\mu}_t^\top \Sigma^{-1} \boldsymbol{r}$. Since this is a scalar, we can write

$$U_{\text{linear}}^2 = (\boldsymbol{\mu}_t^\top \Sigma^{-1} \boldsymbol{r})(\boldsymbol{r}^\top \Sigma^{-1} \boldsymbol{\mu}_t) = \boldsymbol{r}^\top \Sigma^{-1} \boldsymbol{\mu}_t \boldsymbol{\mu}_t^\top \Sigma^{-1} \boldsymbol{r}. \tag{50}$$

To evaluate the expectation, we use the standard result for quadratic forms of Gaussian vectors. For $\boldsymbol{r} \sim \mathcal{N}(\boldsymbol{0}, \Sigma)$ and any symmetric matrix $M$,

$$\mathbb{E}[\boldsymbol{r}^\top M \boldsymbol{r}] = \text{tr}(M\Sigma). \tag{51}$$

Applying this with $M = \Sigma^{-1}\boldsymbol{\mu}_t\boldsymbol{\mu}_t^\top\Sigma^{-1}$:

$$\begin{aligned}
\mathbb{E}[U_{\text{linear}}^2] &= \text{tr}\left(\Sigma^{-1}\boldsymbol{\mu}_t\boldsymbol{\mu}_t^\top\Sigma^{-1}\Sigma\right) \\
&= \text{tr}\left(\Sigma^{-1}\boldsymbol{\mu}_t\boldsymbol{\mu}_t^\top\right) \\
&= \text{tr}\left(\boldsymbol{\mu}_t^\top\Sigma^{-1}\boldsymbol{\mu}_t\right) \quad \text{(cyclic property of trace)} \\
&= \boldsymbol{\mu}_t^\top\Sigma^{-1}\boldsymbol{\mu}_t \quad \text{(trace of a scalar is itself)}.
\end{aligned} \tag{52}$$

Therefore,

$$\boxed{\mathbb{E}[U_{\text{linear}}^2] = \boldsymbol{\mu}_t^\top\Sigma^{-1}\boldsymbol{\mu}_t.} \tag{53}$$

**Quadratic Term Contribution**   We now compute $\mathbb{E}[U_{\text{quadratic}}^2]$. Recall that

$$U_{\text{quadratic}} = \frac{1}{2}Q - B, \quad \text{where } Q := \boldsymbol{r}^\top A \boldsymbol{r}. \tag{54}$$

We first derive the mean and variance of $Q$, then use these to obtain $\mathbb{E}[U_{\text{quadratic}}^2]$.

**Expectation of $Q$.** Using the trace formula for quadratic forms:

$$\mathbb{E}[Q] = \mathbb{E}[\boldsymbol{r}^\top A \boldsymbol{r}] = \operatorname{tr}(A\Sigma). \tag{55}$$

Substituting $A = \Sigma^{-1}\Sigma_t\Sigma^{-1}$:

$$\mathbb{E}[Q] = \operatorname{tr}(\Sigma^{-1}\Sigma_t\Sigma^{-1} \cdot \Sigma) = \operatorname{tr}(\Sigma^{-1}\Sigma_t) = 2B. \tag{56}$$

Therefore, $\mathbb{E}[U_{\text{quadratic}}] = \frac{1}{2}\mathbb{E}[Q] - B = \frac{1}{2}(2B) - B = 0$.

**Variance of $Q$.** To compute $\operatorname{Var}(Q) = \mathbb{E}[Q^2] - (\mathbb{E}[Q])^2$, we need $\mathbb{E}[Q^2]$. Expanding:

$$Q^2 = \left( \sum_{i,j} A_{ij} r_i r_j \right) \left( \sum_{k,l} A_{kl} r_k r_l \right) = \sum_{i,j,k,l} A_{ij} A_{kl} r_i r_j r_k r_l. \tag{57}$$

Taking expectations:

$$\mathbb{E}[Q^2] = \sum_{i,j,k,l} A_{ij} A_{kl} \mathbb{E}[r_i r_j r_k r_l]. \tag{58}$$

For a centered multivariate Gaussian, the fourth-order moments are given by **Isserlis' theorem** (Isserlis, 1918) as

$$\mathbb{E}[r_i r_j r_k r_l] = \Sigma_{ij}\Sigma_{kl} + \Sigma_{ik}\Sigma_{jl} + \Sigma_{il}\Sigma_{jk}. \tag{59}$$

Substituting this into $\mathbb{E}[Q^2]$ yields three terms:

$$\mathbb{E}[Q^2] = T_1 + T_2 + T_3, \tag{60}$$

where

$$T_1 := \sum_{i,j,k,l} A_{ij} A_{kl} \Sigma_{ij} \Sigma_{kl}, \tag{61}$$

$$T_2 := \sum_{i,j,k,l} A_{ij} A_{kl} \Sigma_{ik} \Sigma_{jl}, \tag{62}$$

$$T_3 := \sum_{i,j,k,l} A_{ij} A_{kl} \Sigma_{il} \Sigma_{jk}. \tag{63}$$

**Evaluation of $T_1$.** This term factorizes:

$$T_1 = \left( \sum_{i,j} A_{ij} \Sigma_{ij} \right) \left( \sum_{k,l} A_{kl} \Sigma_{kl} \right) = (\operatorname{tr}(A\Sigma))^2 = (\mathbb{E}[Q])^2. \tag{64}$$

**Evaluation of $T_2$.** Define $C := A\Sigma = \Sigma^{-1}\Sigma_t$. Then:

$$
\begin{aligned}
T_2 &= \sum_{i,j,k,l} A_{ij} A_{kl} \Sigma_{ik} \Sigma_{jl} \\
&= \sum_{i,k} \left( \sum_j A_{ij} \Sigma_{jl} \right) \left( \sum_l A_{kl} \Sigma_{ik} \right) \\
&= \sum_{i,k} C_{il} C_{ki} \quad \text{(where we sum over } j \text{ and } l) \\
&= \operatorname{tr}(C^2) = \operatorname{tr}\left((A\Sigma)^2\right) = \operatorname{tr}\left((\Sigma^{-1}\Sigma_t)^2\right).
\end{aligned}
\tag{65}
$$

**Evaluation of $T_3$.** By a similar index manipulation (using symmetry of $A$ and $\Sigma$):

$$T_3 = \sum_{i,j,k,l} A_{ij} A_{kl} \Sigma_{il} \Sigma_{jk} = \operatorname{tr}\left((A\Sigma)^2\right) = T_2. \tag{66}$$

**Combining the terms.** We obtain:

$$\mathbb{E}[Q^2] = T_1 + T_2 + T_3 = (\mathbb{E}[Q])^2 + 2\operatorname{tr}\left((\Sigma^{-1}\Sigma_t)^2\right). \tag{67}$$

Therefore, the variance of $Q$ is:

$$\operatorname{Var}(Q) = \mathbb{E}[Q^2] - (\mathbb{E}[Q])^2 = 2\operatorname{tr}\left((\Sigma^{-1}\Sigma_t)^2\right). \tag{68}$$

**Back to $U_{\text{quadratic}}$.** Since $U_{\text{quadratic}} = \frac{1}{2}Q - B$ and $B$ is a constant:

$$\operatorname{Var}(U_{\text{quadratic}}) = \operatorname{Var}\left(\frac{1}{2}Q\right) = \frac{1}{4}\operatorname{Var}(Q) = \frac{1}{2}\operatorname{tr}\left((\Sigma^{-1}\Sigma_t)^2\right). \tag{69}$$

Since $\mathbb{E}[U_{\text{quadratic}}] = 0$, we have:

$$\boxed{\mathbb{E}[U_{\text{quadratic}}^2] = \operatorname{Var}(U_{\text{quadratic}}) = \frac{1}{2}\operatorname{tr}\left((\Sigma^{-1}\Sigma_t)^2\right).} \tag{70}$$

### A.2.3. FINAL RESULT: CLOSED-FORM FISHER INFORMATION

Combining all three contributions:

$$I(\boldsymbol{p}, t) = \mathbb{E}[U_{\text{linear}}^2] + 2\mathbb{E}[U_{\text{linear}}U_{\text{quadratic}}] + \mathbb{E}[U_{\text{quadratic}}^2]$$

$$= \boldsymbol{\mu}_t^\top \Sigma^{-1} \boldsymbol{\mu}_t + 0 + \frac{1}{2}\operatorname{tr}\left((\Sigma^{-1}\Sigma_t)^2\right). \tag{71}$$

---

**Closed-Form Fisher Information**

$$I(\boldsymbol{p}, t) = \underbrace{\boldsymbol{\mu}_t^\top \Sigma^{-1} \boldsymbol{\mu}_t}_{I_\mu:\,\text{Mean Term}} + \underbrace{\frac{1}{2}\operatorname{tr}\left((\Sigma^{-1}\Sigma_t)^2\right)}_{I_\Sigma:\,\text{Covariance Term}} \tag{72}$$

---

### A.2.4. PROOF OF THE CRAMÉR–RAO LOWER BOUND

**Theorem A.1** (Cramér–Rao Lower Bound). *The variance of the intrinsic time $\tau$ conditioned on template location $\boldsymbol{p}$ and chronological time $t$ satisfies*

$$\operatorname{Var}_{p(\boldsymbol{d}|\boldsymbol{p},t)}(\tau) \geq \frac{1}{I(\boldsymbol{p}, t)}. \tag{73}$$

*Proof.* Let $U(\boldsymbol{d}; \boldsymbol{p}, t) := \partial_t \log p(\boldsymbol{d}|\boldsymbol{p}, t)$ denote the *score function*, i.e., the derivative of the log-likelihood with respect to the temporal parameter $t$. The proof proceeds in four steps.

**Step 1: Score has zero mean.** Under standard regularity conditions,

$$\mathbb{E}_{p(\boldsymbol{d}|\boldsymbol{p},t)}[U(\boldsymbol{d}; \boldsymbol{p}, t)] = \int \frac{\partial_t p(\boldsymbol{d} \mid \boldsymbol{p}, t)}{p(\boldsymbol{d} \mid \boldsymbol{p}, t)} p(\boldsymbol{d} \mid \boldsymbol{p}, t)\, d\boldsymbol{d} = \partial_t \int p(\boldsymbol{d} \mid \boldsymbol{p}, t)\, d\boldsymbol{d} = \partial_t 1 = 0. \tag{74}$$

**Step 2: Unbiasedness condition.** We assume $\mathbb{E}_{p(\boldsymbol{d}|\boldsymbol{p},t)}[\tau] = t$, meaning that the population-average intrinsic time equals the chronological time. Differentiating both sides with respect to $t$:

$$1 = \partial_t \mathbb{E}_{p(\boldsymbol{d}|\boldsymbol{p},t)}[\tau] = \int \tau \cdot \partial_t p(\boldsymbol{d} \mid \boldsymbol{p}, t)\, d\boldsymbol{d} = \int \tau \cdot p(\boldsymbol{d} \mid \boldsymbol{p}, t) \cdot U(\boldsymbol{d}; \boldsymbol{p}, t)\, d\boldsymbol{d} = \mathbb{E}_{p(\boldsymbol{d}|\boldsymbol{p},t)}[\tau \cdot U]. \tag{75}$$

**Step 3: Covariance identity.** Since $\mathbb{E}_{p(\boldsymbol{d}|\boldsymbol{p},t)}[\tau] = t$ and $\mathbb{E}_{p(\boldsymbol{d}|\boldsymbol{p},t)}[U] = 0$, we have

$$\operatorname{Cov}_{p(\boldsymbol{d}|\boldsymbol{p},t)}(\tau, U) = \mathbb{E}_{p(\boldsymbol{d}|\boldsymbol{p},t)}[(\tau - t)(U - 0)] = \mathbb{E}_{p(\boldsymbol{d}|\boldsymbol{p},t)}[\tau \cdot U] - t \cdot \mathbb{E}_{p(\boldsymbol{d}|\boldsymbol{p},t)}[U] = 1. \tag{76}$$

**Step 4: Cauchy–Schwarz inequality.** Applying the Cauchy–Schwarz inequality:

$$1 = |\operatorname{Cov}_{p(\boldsymbol{d}|\boldsymbol{p},t)}(\tau, U)|^2 \leq \operatorname{Var}_{p(\boldsymbol{d}|\boldsymbol{p},t)}(\tau) \cdot \operatorname{Var}_{p(\boldsymbol{d}|\boldsymbol{p},t)}(U) = \operatorname{Var}_{p(\boldsymbol{d}|\boldsymbol{p},t)}(\tau) \cdot I(\boldsymbol{p}, t). \tag{77}$$

Rearranging yields $\operatorname{Var}_{p(\boldsymbol{d}|\boldsymbol{p},t)}(\tau) \geq 1/I(\boldsymbol{p}, t)$. $\qquad\square$

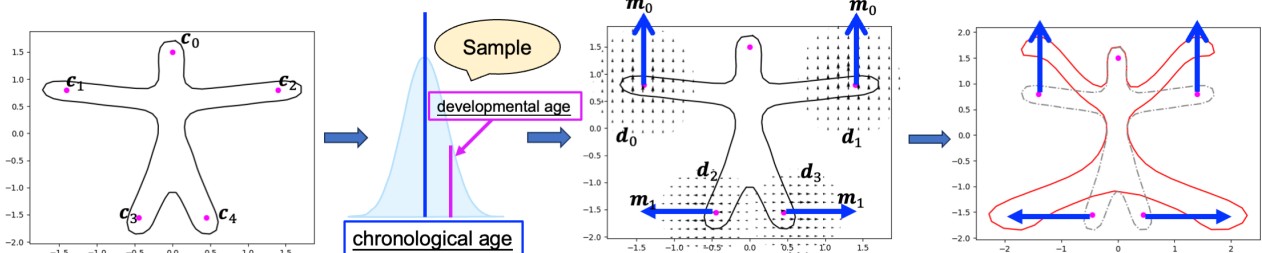

*Figure 6.* Starman dataset generation. **Left:** Template shape with four control points governing arm and leg deformations. **Middle:** Shape evolution across physical time $t \in [0, 1]$. **Right:** Temporal uncertainty functions $\sigma_\tau(t)$ for Starman(L), where arms (red) exhibit early growth and legs (blue) exhibit late growth.

### A.2.5. GEOMETRIC INTERPRETATION OF THE FISHER INFORMATION

The additive structure of the Fisher Information (Eq. (72)) admits an elegant geometric interpretation. The total information $I(\boldsymbol{p}, t)$ decomposes into contributions from two orthogonal sub-manifolds (Skovgaard, 1984; Amari, 2016):

**Manifold of Means ($I_\mu = \boldsymbol{\mu}_t^\top \Sigma^{-1} \boldsymbol{\mu}_t$).** This term is the squared Mahalanobis norm of the mean velocity $\boldsymbol{\mu}_t$, measuring the squared rate of shape evolution on the statistical manifold of Gaussian means. The precision matrix $\Sigma^{-1}$ serves as the Riemannian metric, weighting structural changes by the local signal-to-noise ratio. A small displacement in a high-precision (low-variance) region contributes more information than a large displacement in a noisy region.

**SPD Manifold ($I_\Sigma = \frac{1}{2}\text{tr}((\Sigma^{-1}\Sigma_t)^2)$).** This term is the squared velocity on the manifold of symmetric positive definite (SPD) matrices, quantifying the squared rate of uncertainty evolution. It reveals that changes in the dispersion profile—independent of any mean shift—constitute an orthogonal source of temporal evidence.

This decomposition justifies our focus on $I_\mu$ for developmental time estimation: while $I_\Sigma$ is statistically informative, it captures how population diversity evolves rather than how individual subjects progress along the mean trajectory.

## B. Datasets and Experiments

### B.1. Starman Dataset

We generate two synthetic 2D datasets with known ground-truth intrinsic time to validate `PRISM` under controlled conditions: **Starman(G)** with global (shape-level) temporal uncertainty, and **Starman(L)** with local (spatially-varying) temporal uncertainty. As illustrated in Fig. 6, each starman shape is synthesized by applying a covariate-controlled deformation to a template shape, representing different poses.

**Shape Parameterization.** The template is a closed polygon with 100 uniformly sampled vertices. Four control points $\{\boldsymbol{c}_i\}_{i=0}^3$ govern limb deformations via Gaussian radial basis functions:

$$\boldsymbol{d}_i(\boldsymbol{p}, \tau_i) = \tau_i \cdot \exp\left(-\frac{\|\boldsymbol{p} - \boldsymbol{c}_i\|^2}{2\sigma^2}\right) \cdot \boldsymbol{v}_i, \quad \sigma = 0.5 \tag{78}$$

where $\tau_i$ is the intrinsic time for control point $i$, and $\boldsymbol{v}_i$ is the deformation direction (vertical for arms $i \in \{0, 1\}$; horizontal for legs $i \in \{2, 3\}$). The total deformation is $\boldsymbol{d}(\boldsymbol{p}) = \sum_{i=0}^3 \boldsymbol{d}_i(\boldsymbol{p}, \tau_i)$.

**Temporal Uncertainty Model.** To simulate inter-subject developmental variability, we model intrinsic time $\tau$ as a function of physical time $t$ and a subject-specific latent variable $z \sim \mathcal{N}(0, 1)$:

$$\tau = t + z \cdot \sigma_\tau(t) \tag{79}$$

The temporal uncertainty $\sigma_\tau(t)$ follows a logistic function capturing age-dependent heteroscedasticity:

$$\sigma_\tau(t) = \sigma_{\min} + \frac{\sigma_{\max} - \sigma_{\min}}{1 + \exp(-(t - t_{50})/k)} \tag{80}$$

**Starman(G): Global Temporal Uncertainty.** All control points share a single intrinsic time $\tau_0 = \tau_1 = \tau_2 = \tau_3 = \tau$, with parameters $(\sigma_{\min}, \sigma_{\max}, t_{50}, k) = (0.01, 0.20, 0.88, 0.12)$. This models uniform developmental timing across the entire body.

**Starman(L): Local Temporal Uncertainty.** Arms and legs follow distinct developmental trajectories with separate intrinsic times:

$$
\begin{aligned}
\tau_{\text{arm}} &= t + z \cdot \sigma_{\tau,\text{arm}}(t), \quad (\sigma_{\max}, t_{50}, k) = (0.15, 0.30, 0.10) \\
\tau_{\text{leg}} &= t + z \cdot \sigma_{\tau,\text{leg}}(t), \quad (\sigma_{\max}, t_{50}, k) = (0.20, 0.88, 0.12)
\end{aligned}
\tag{81}
$$

**Dataset Statistics.** Each dataset contains 1,000 training and 1,000 testing subjects, with 1–9 longitudinal observations per subject sampled uniformly across $t \in [0, 1]$.

## B.2. ANNY Dataset

We use ANNY (Brégier et al., 2025), a parametric 3D human body model spanning the full lifespan from infancy to old age. ANNY captures the morphological changes during childhood development, including evolving body proportions and limb-to-torso ratios.

**Shape Generation.** Each 3D mesh is generated by the ANNY model with age as the sole factor influencing the shapes. The output is a triangulated mesh.

**Temporal Uncertainty Model.** We simulate inter-subject temporal variability following a heteroscedastic model for bone age uncertainty (Cole et al., 2010; Thodberg et al., 2008), where temporal uncertainty increases during puberty, as

$$
\tau = t + z \cdot \sigma_\tau(t), \quad z \sim \mathcal{N}(0, 1),
\tag{82}
$$

whose temporal uncertainty function follows a sigmoid:

$$
\sigma_\tau(t) = \sigma_{\min} + \frac{\sigma_{\max} - \sigma_{\min}}{1 + \exp(-k(t - t_{\text{pub}}))}
\tag{83}
$$

where $t \in [0, 20]$ years is physical age, $\sigma_{\min} = 0.4$ years, $\sigma_{\max} = 1.3$ years, $t_{\text{pub}} = 8$ years marks puberty onset, and $k = 0.5$ controls transition sharpness. We choose these parameters to reflect general trends in pediatric growth literature (Cole et al., 2010; Thodberg et al., 2008).

**Dataset Statistics.** We generate 1,000 training subjects and 100 testing subjects. The training set contains 90% cross-sectional subjects and 10% longitudinal subjects. The test set includes 50 cross-sectional and 50 longitudinal subjects.

## B.3. Pediatric Airway Dataset

Our pediatric airway dataset comprises 358 CT scans from 264 subjects, with ages ranging from 0 to 19.4 years. Some subjects have multiple longitudinal scans (up to 11 scans per subject), providing valuable temporal information for modeling airway development. We perform an 80/20 train-test split at the subject level to prevent data leakage.

### B.3.1. DATA ACQUISITION

The pediatric airway dataset consists of CT scans from children aged 1 month to 19 years. The dataset includes 230 single-visit subjects and 34 subjects with repeated imaging (2–11 scans per subject).

### B.3.2. PREPROCESSING PIPELINE

**Airway Segmentation.**  We employ a two-stage deep learning approach for automatic airway segmentation. The first stage uses a coarse-resolution UNet (Çiçek et al., 2016) to generate an initial prediction, which then guides a second full-resolution UNet for refined segmentation. The segmentation model was trained on 68 manually annotated CT-segmentation pairs.

**Centerline Extraction.**  Following (Hong et al., 2013), we extract the airway centerline by solving Laplace's equation within the segmented volume. The centerline is defined as the locus of centers of heat-value isosurfaces.

**Airway Straightening.**  The pediatric airway exhibits significant anatomical curvature from the nasal cavity to the carina. To disentangle shape variation from pose variation, we apply a centerline-based straightening transformation using a rotation

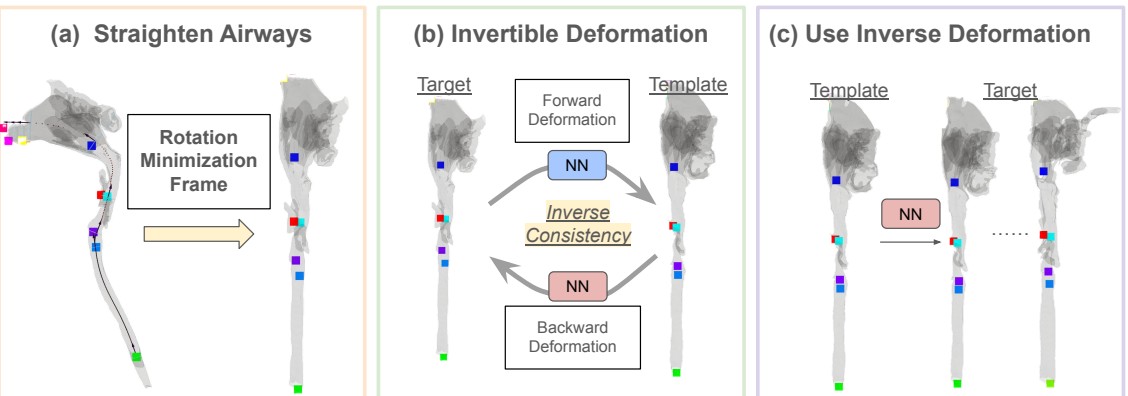

*Figure 7.* Preprocessing pipeline for pediatric airway shape analysis. (a) Raw airway geometries are straightened using a rotation-minimizing frame to remove extrinsic pose variations while preserving intrinsic shape characteristics. (b) An invertible neural network learns bidirectional deformations between each target shape and a common template, with inverse consistency regularization ensuring geometric coherence. (c) The learned inverse deformation is applied to map the template onto each target, establishing dense point correspondences across the population for downstream developmental modeling.

minimizing frame (RMF) ([Bishop, 1975](#)) constructed along the extracted airway centerline. As shown in Fig. 7(a), this RMF maps the curved airway into a cylinder-like representation where the $z$-axis corresponds to airway depth and the $xy$-plane captures cross-sectional variations.

**SDF Sampling.**    Following ([Park et al., 2019](#); [Sitzmann et al., 2020](#)), we compute signed distance function (SDF) samples for each straightened airway mesh. On-surface points are sampled with corresponding vertex normals; off-surface points are sampled within a padded bounding box with SDF values computed via nearest-surface distance.

**Out-of-Distribution Cases**    We exclude 31 scans with subglottic stenosis from training. Subglottic stenosis is a pathological narrowing of the airway below the vocal cords, resulting in abnormal cross-sectional geometry. These cases serve as out-of-distribution (OOD) samples for evaluating anomaly detection capabilities.

### B.4. Baseline Adaption for OOD

We adapt A-SDF and NAISR for OOD detection via test-time optimization. Given a test shape, we freeze the network and optimize the latent time $\hat{\tau}$ to minimize reconstruction loss. The anomaly score is defined as $\hat{\tau} - t$, measuring the deviation between optimized intrinsic time and chronological age. This formulation is clinically motivated: airway obstruction (e.g., subglottic stenosis) manifests as abnormal narrowing, causing the affected region to resemble a developmentally younger airway. A negative score thus indicates pathological underdevelopment. For PRISM (Global), we apply the same strategy using the mean predicted time as Eq. (15). For PRISM (Local), we use the spatially-varying OOD score using Eq. (20).

### B.5. Additional Experiment Results

As another qualitative evaluation, Fig. 8 shows PRISM applied to the pediatric airway dataset. For a queried chronological age $t$, PRISM infers a distribution over the intrinsic (biological) time $\tau$, summarized by its mean $t$ and standard deviation $\sigma_\tau$. Propagating this temporal uncertainty through the shape evolution $\mu(\cdot)$, we render the central anatomy $\mu(\boldsymbol{p}, t)$ together with the shapes obtained at the $t \pm 2\sigma_\tau$ endpoints of the intrinsic-time distribution, which delineate the corresponding geometric envelope.

Three aspects are worth noting. First, the mean trajectory recovers a smooth, non-linear progression of anatomical growth consistent with expected airway development. Second, most observed airway shapes fall within the geometric envelope spanned by the $t \pm 2\sigma_\tau$ reconstructions, indicating that the propagated temporal uncertainty plausibly covers the range of real anatomical variation. Third, although exact temporal matches to a queried age $t$ are rarely available under sparse clinical acquisition, the reconstruction remains in close morphological agreement with the nearest observed anatomy, suggesting

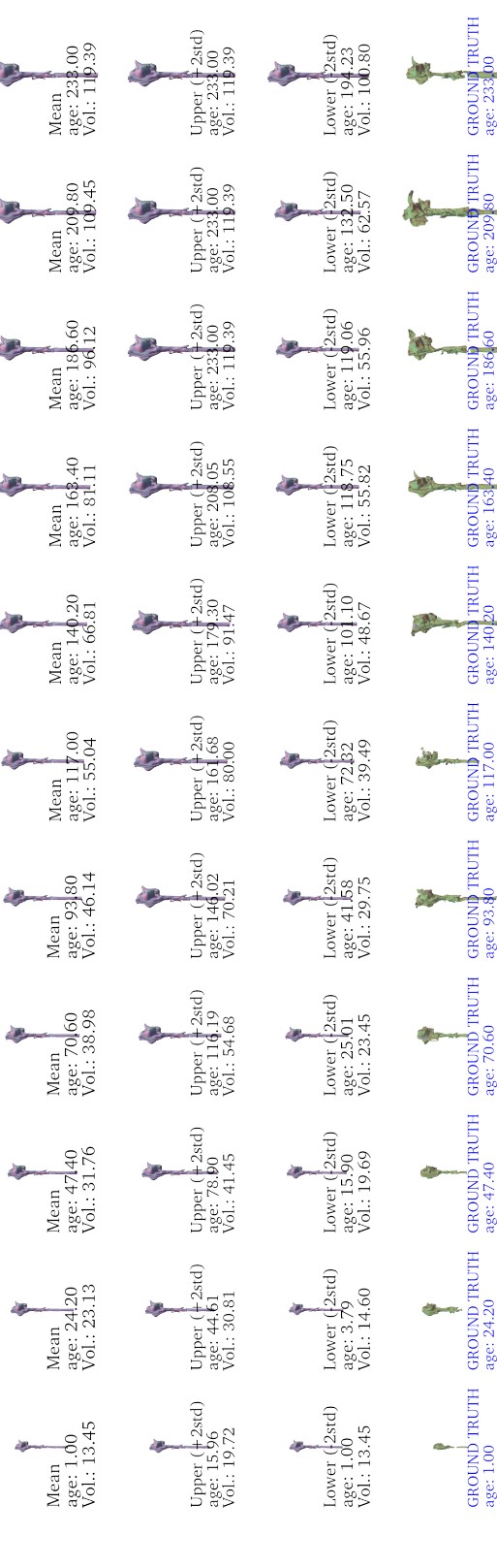

*Figure 8.* Probabilistic reconstruction of longitudinal pediatric airway development. This visualization demonstrates PRISM's capacity to represent real-world clinical data. The columns correspond to increasing ages from left to right. (Row 1) Mean Trajectory: The predicted mean shape evolution ($\mu(\boldsymbol{p}, t)$), capturing the central trend of anatomical growth. (Rows 2 & 3) Uncertainty Bounds: The shapes generated at the upper ($t + 2\sigma_\tau$) and lower ($t - 2\sigma_\tau$) bounds of the predicted intrinsic time distribution. These bounds visualize the learned population bands. (Bottom Row) Nearest Ground Truth: The observed anatomy from the subject with the age closest to the queried time $t$. Note that due to the sparsity of clinical acquisition, exact temporal matches are unavailable; the close morphological alignment despite this temporal gap highlights the model's robustness in capturing developmental trends. The correspondence between the probabilistic reconstruction and the ground truth validates the model's ability to capture complex, non-linear developmental patterns and their intrinsic biological variability.

that `PRISM` interpolates population-level developmental trends rather than memorizing individual training subjects. Taken together, these qualitative results support our claim that `PRISM` successfully captures the biological variability of the airway population.

### B.6. Architectural Comparison with NAISR and A-SDF

*Table 7.* Architectural and modeling comparison of PRISM with its closest counterparts. Parameter counts are for the *trainable* core model $f(\cdot)$ under comparison. Per-shape latent deformation codes are excluded for all methods, as their count scales with dataset size rather than model capacity. PRISM's amortized inverse encoder is likewise excluded.

| Method | Deformable | Intrinsic time | Supervision | #Params |
|---|---|---|---|---|
| A-SDF | ✗ | global | SDF field | 1.97M |
| NAISR | ✓ | global | SDF field | 3.68M |
| PRISM (ours) | ✓ | **local** | **deformations** | **3.57M** |

Table 7 summarizes the key differences between `PRISM` and its two closest counterparts. **(i) Representation.** Both `PRISM` and NAISR are deformation-based, representing each shape as a deformation of a shared template, whereas A-SDF conditions an implicit decoder on a latent code without an explicit deformation field. This deformation-based formulation enables principled extrapolation along the developmental axis and underlies `PRISM`'s advantage in longitudinal prediction, as shown in Tab. 4. `PRISM` and NAISR also differ in how deformation is parameterized. NAISR represents each shape by warping the query point into a shared template, $S_i(\boldsymbol{p}) = T(\boldsymbol{p} + d_i(\boldsymbol{p}))$, i.e. the deformation $d_i$ maps points *into* the template frame. Inverting this map to locate a *fixed* template point on each subject is not modeled, so a common template point cannot be tracked across subjects to define per-point quantities. `PRISM` instead learns the forward deformation $\boldsymbol{q} \mapsto \boldsymbol{q} + D_i(\boldsymbol{q})$ that places each fixed template point $\boldsymbol{q}$ on subject $i$, providing the cross-subject correspondence required to define spatially varying $\hat{\tau}$ and $\Sigma$. **(ii) Intrinsic time.** `PRISM` estimates developmental time *locally*, at each spatial location, rather than as a single global value per shape as in NAISR and A-SDF. This locality is precisely what makes spatially-resolved out-of-distribution (OOD) detection meaningful, since a localized anomaly cannot be detected from a single global time estimate. **(iii) Supervision signal.** NAISR and A-SDF are supervised on the reconstructed SDF field, whereas `PRISM` is supervised directly on the (probabilistic) deformation field, predicting $(\mu, \Sigma)$ rather than a point SDF value. `PRISM` therefore receives its learning signal directly, without routing it through SDF reconstruction, and yields calibrated uncertainty as a native output.

