# OpenReview forum: "$\texttt{PRISM}$:A 3D Probabilistic Neural Representation for Interpretable Shape Modeling"
_ICML.cc/2026/Conference — ICML 2026 regular_

### Official Review · Reviewer_nfhK · 2026-03-10

**Soundness:** 3
**Presentation:** 2
**Significance:** 3
**Originality:** 3
**Overall Recommendation:** 4
**Confidence:** 5

**Summary:**

This paper introduces PRISM, a framework for modeling covariate-dependent anatomical shape evolution with spatially resolved uncertainty quantification. The method integrates implicit neural representations with statistical shape analysis to model the conditional distribution of shapes given developmental or clinical covariates. Unlike traditional global time-warping approaches, PRISM aims to capture spatially heterogeneous dynamics and provide continuous estimates of both the population mean shape and covariate-dependent uncertainty at arbitrary spatial locations.

A central theoretical contribution is a closed-form Fisher Information metric that enables analytically tractable, local temporal uncertainty quantification via automatic differentiation. The framework is evaluated on three synthetic datasets and one clinical airway dataset. The authors report strong performance across tasks and emphasize interpretability and clinically meaningful uncertainty estimation. Code is promised for public release.

**Compliance With Llm Reviewing Policy:**

Affirmed.

**Final Justification:**

The authors clarified my main issues in their rebuttal.

**Key Questions For Authors:**

1. Robustness on Real Clinical Data
Performance on the airway dataset appears less consistent than on synthetic datasets. Can you clarify in which regimes PRISM underperforms and whether this is due to model bias, limited sample size, or data noise?
If the method can be shown to remain stable under realistic clinical variability, this would substantially strengthen the empirical contribution.

2. OOD Detection Protocol and Thresholding
How is the OOD threshold selected? Is zero used implicitly, or is the threshold optimized on a validation set? How many samples are OOD versus in-distribution, and can you report PPV and NPV in addition to sensitivity/specificity?
Clear reporting of these details could materially affect the evaluation of PRISM’s reliability in safety-critical applications.

3. Interpretation of the Fisher-Based Uncertainty Metric
How should the Fisher Information–derived local temporal uncertainty be interpreted clinically? Does it correlate with known developmental variability or measurement noise in real data?
Empirical validation of this interpretability claim would significantly strengthen the practical impact of the theoretical contribution.

**Limitations:**

Yes

**Strengths And Weaknesses:**

Strengths
1.	Timely and Clinically Relevant Problem
Modeling covariate-driven anatomical shape evolution with spatially varying uncertainty is an important problem in medical imaging and computational anatomy. Moving beyond global time-warping formulations is well-motivated.
2.	Integration of Implicit Representations and Statistical Modeling
Bridging implicit neural representations with uncertainty-aware statistical shape analysis is conceptually elegant. The formulation allows spatially continuous predictions rather than mesh-bound or discretized outputs.
3.	Theoretical Contribution via Fisher Information Metric
The derivation of a closed-form Fisher Information metric for local temporal uncertainty quantification is a technically meaningful contribution. Enabling efficient computation via automatic differentiation strengthens the methodological rigor.
4.	Unified Framework Across Tasks
The method is demonstrated across multiple synthetic datasets and one real clinical dataset, suggesting generality. The emphasis on interpretable uncertainty estimates is a practical advantage.

Weaknesses
1.	Inconsistent Empirical Results
Performance on the airway clinical dataset appears mixed relative to synthetic experiments. This raises concerns about robustness and generalizability in real-world settings, where anatomical variability and noise are more complex.
2.	Omission of OOD Detection Details
The manuscript lacks clarity regarding the out-of-distribution (OOD) detection setup. It is unclear whether a zero threshold is used, whether the threshold is optimized, and how many cases are in-distribution versus OOD. Without reporting metrics such as positive predictive value (PPV) and negative predictive value (NPV), the practical reliability of OOD detection cannot be properly assessed.
3.	Presentation and Formatting Issues
The manuscript exhibits inconsistent line spacing that does not comply with standard conference formatting guidelines. Additionally, Appendix B1 contains a figure that is not properly referenced in the main text. These issues detract from the overall polish expected at a top-tier venue.
4.	Limited Clinical Validation
Only one clinical dataset is included. Given the clinical framing of the contribution, broader validation across additional real datasets would strengthen claims of general applicability.

---

> ### Author Rebuttal · Authors · 2026-03-30
>
> We thank the reviewer for the thorough evaluation and constructive questions.
>
> ## Weaknesses
>
> > W1: Inconsistent empirical results.
>
> We note that on synthetic data, ground-truth intrinsic time is available and PRISM recovers it accurately (Tab. 1). On clinical data, no such ground truth exists — chronological age serves as a proxy, and the performance gap stems from the amortized inverse encoder, not the probabilistic model. We conducted a test-time optimization (TTO) ablation on the airway dataset, i.e., freezing the trained model and optimizing $\tau$ per test case, which is the same inference protocol used by NAISR and A-SDF:
>
> *Intrinsic Time Estimation (Airway):*
>
> | Method | Inference | r ↑ | R² ↑ | MAE (mo.) ↓ | Time/case |
> | :---- | :---- | :---- | :---- | :---- | :---- |
> | A-SDF | TTO | 0.745 | 0.334 | 42.886 | 16.86s |
> | NAISR | TTO | 0.908 | 0.820 | 20.512 | 24.36s |
> | PRISM | Amortized | 0.881 | 0.770 | 23.415 | 0.48s |
> | **PRISM** | **TTO** | **0.925** | **0.855** | **18.794** | **8.28s** |
>
> PRISM(TTO) achieves the best performance across all metrics while being 3× faster than NAISR(TTO). This improvement propagates to downstream tasks — personalized prediction with TTO-estimated $\hat{\tau}$ also yields the best results:
>
> *Personalized Prediction (Airway):*
>
> | Method | Inference | CD ↓ | HD ↓ | EMD ↓ |
> | :---- | :---- | :---- | :---- | :---- |
> | A-SDF | TTO | 0.123 | 10.762 | 2.034 |
> | NAISR | TTO | 0.065 | 9.599 | 1.341 |
> | PRISM | Amortized | 0.088 | 12.735 | 1.347 |
> | **PRISM** | **TTO** | **0.064** | **9.031** | **1.264** |
>
> These results indicate that PRISM's probabilistic model outperforms all baselines on clinical data. The gap in Tab. 2 reflects an accuracy–efficiency trade-off: the amortized encoder provides 17× faster inference at the cost of marginal accuracy reductions.
>
>
> > W2: Omission of OOD detection details.
>
> We appreciate this question. The OOD setup is described in Sec. 5.1.1 and Appendix B.5, but we agree that some protocol details were missing. The setup uses 137 in-distribution (ID) healthy test cases vs 31 out-of-distribution (OOD) subglottic stenosis cases, with Youden's J statistic for threshold selection. At the optimal operating point, PRISM(Local) achieves PPV=0.510 and NPV=0.950. The NPV is clinically relevant: when the model identifies a case as normal, it is correct 95% of the time. The lower PPV reflects class imbalance (31 OOD vs 137 ID).
>
> > W3: Presentation and formatting issues.
>
> We appreciate the careful reading and will fix all formatting issues (line spacing, AUTHORERR tag, unreferenced Appendix B1 figure) in the camera-ready.
>
> > W4: Limited clinical validation.
>
> We acknowledge that only one clinical dataset is included. However, we evaluate PRISM across multiple complementary tasks on this dataset — reconstruction, time estimation, personalized prediction, and OOD detection — each providing independent evidence for the framework's validity. The dataset itself is also non-trivial: 358 CT scans from 264 subjects (ages 0–19 years), including 34 with longitudinal sequences and 31 pathological cases. Broader validation across additional anatomies is an important direction for future work.
>
>
> ## Key Questions
>
> > Q1: Robustness on real clinical data.
>
> Please see W1 above. In brief: (1) no ground-truth intrinsic time exists for clinical data; (2) PRISM(TTO) outperforms all baselines; (3) the amortized inverse encoder trades marginal accuracy for 17× faster inference.
>
> > Q2: OOD detection protocol and thresholding.
>
> Please see W2 above.
>
> > Q3: Interpretation of the Fisher-based uncertainty metric.
>
> The Fisher Information-derived temporal uncertainty quantifies aleatoric population variability (i.e., the inherent uncertainty in the data distribution rather than model output variability) in the developmental stage, i.e., $p(\tau \mid \mathbf{p},t)$. For example, among all 3-year-old children, some childrens’ airways at a given region may appear more similar to the airways of  4-year-old children, while others resemble more closely 2-year-old children. A high temporal uncertainty at that location reflects this large developmental variation within the population.
>
> We provide empirical validation at two levels. First, on synthetic datasets — which contain no measurement noise, only the designed developmental uncertainty — PRISM accurately recovers the true conditional distribution $p(\tau \mid \boldsymbol{p},t)$, and the decoded shapes at $\mu \pm 2\sigma$ match ground truth (Fig. 2). This confirms that the metric captures genuine temporal variability, not measurement artifacts. In clinical data, ground-truth uncertainty is unavailable, but anatomical knowledge provides validation: the learned uncertainty (Fig. 4\) shows that soft-tissue regions (e.g., base of tongue) exhibit wider uncertainty bands than more rigid structures (e.g., carina), consistent with Zhou et al. (2021). These results confirm the clinical interpretability of our FI-based uncertainty metric.

---

> > ### Author Rebuttal · Reviewer_nfhK · 2026-04-03
> >
> > Thank you for the rebuttal. It has fully resolved my comments.

---

### Official Review · Reviewer_r9XV · 2026-03-11

**Soundness:** 3
**Presentation:** 2
**Significance:** 3
**Originality:** 3
**Overall Recommendation:** 5
**Confidence:** 3

**Summary:**

PRISM introduces a novel uncertainty modeling framework for statistical shape population representation on conditional variables such as time. The primary contribution of the paper are the modeling of a spatially varying, uncertainty-aware implicit neural field, the derivation of the closed-form Fisher-Information Metric which provides tractable uncertainty quantification and lastly the introduction of an architectural component, i.e. the specifically, an inverse encoder which allows to estimate correlates from the implicit shape model. The proposed method outperforms selected SOTA methods in different tasks.

**Compliance With Llm Reviewing Policy:**

Affirmed.

**Final Justification:**

Initially, I had some reservations regarding the presentation of the paper, but not regarding the method and/or significance of the results, as some things were unclear.  The authors have done a great job of answering my questions and concerns in the rebuttal, which has clarified details I was previously missing. This has consequently led me to increase my score from 4->5, which remains unchanged in my final justification. I look forward to the incorporation of the clarifications in the final manuscript.

**Key Questions For Authors:**

1. Could you please clearly state architectural and modeling differences between NAISR, A-SDF and PRISM in the main manuscript? This would help me to evaluate which aspects of PRISM lead it to (clearly) outperform its counterparts (A-SDF, NAISR)? I am aware that the conceptual and theoretical contribution is clearly strong, but I would like to ensure that the compairison is fair (eg model footprint). I apogize if I have missed these important details, and will reconsider my score during the discussion phase.

2. Which role do regularization terms play for the uncertainty modeling framework? Do results drop (dramatically) without e.g Eikonal constraints? I would appreciate an ablation in case this does change performance?

3. Would smoothness constraints such as Lipschitz reg c.f. [1] help?

[1] Liu HT, Williams F, Jacobson A, Fidler S, Litany O. Learning smooth neural functions via lipschitz regularization. InACM SIGGRAPH 2022 conference proceedings 2022 Jul 27 (pp. 1-13).

**Limitations:**

yes

**Strengths And Weaknesses:**

#### Strengths:

- The theoretical contribution of the paper, i.e. the framing of the statistical shape population model, and the derivation of the closed-form Fisher-Information Metric seem theoretically sound and genuinely novel and interesting. I am not aware of any such method within the context of implicit shape modeling. In this context, I would assess it as novel, interesting and of relevance to the community which has been recently embracing INRs as a population modeling framework (both imaging and shapes).
- The proposed method consistently outperforms appropriate baseline methods, some with high margins.

#### Weaknesses

- The paper is not particularly easy to follow for people coming from an implicit shape modeling background and INR literature. Particularly, the utilized architecture and its difference to baseline methods (NAISR, A-SDF) is not (clearly) evident from the main manuscript, making it difficult to assess which aspects allow PRISM to clearly outperform in tasks such as Table 4 and Table 5. For instance, the reader needs to check section A1 to discover that the approach uses hyper-networks, and it is not evident why the authors select this specific architectural setup. Another aspect here is that the authors do not introduce signed distance fields (please correct me if I am wrong), and the usage of e.g. Eikonal Constraints comes as a surprise in the Appendix.
- It is thus not clear if the improvements come primarily from architectural differences, or the problem formulation. This is particularly evident in Sec 5.1.2 which briefly introduces baselines, but doesn't go into the differences.
- Given that the proposed framework leverages implicit neural representations, it (1) lacks comparison in related work to the conditional shape modeling and deformation field literature in INRs e.g. [1-4], and correlate analysis works such as [3]. I find [3] to be particularly relevant given that it uses a conditional architecture and PCA to estimate correlates. It would also be nice to include some adjacent uncertainty literature in INRs [5,6], even if they are not well cited.

[1] Atzmon, Matan, et al. "Augmenting implicit neural shape representations with explicit deformation fields." arXiv preprint arXiv:2108.08931 (2021).

[2] DANNECKER, Maik, et al. CINeMA: Conditional Implicit Neural Multi-Modal Atlas for a Spatio-Temporal Representation of the Perinatal Brain. IEEE Transactions on Medical Imaging, 2025.

[3] DANNECKER, Maik, et al. Cina: Conditional implicit neural atlas for spatio-temporal representation of fetal brains. In: International Conference on Medical Image Computing and Computer-Assisted Intervention. Cham: Springer Nature Switzerland, 2024. S. 181-191.

[4] GROßBRÖHMER, Christoph, et al. SINA: Sharp implicit neural atlases by joint optimisation of representation and deformation. In: International Workshop on Biomedical Image Registration. Cham: Springer Nature Switzerland, 2024. S. 165-180.

[5] SAKLANI, Shanu, et al. Uncertainty-informed volume visualization using implicit neural representation. In: 2024 IEEE Workshop on Uncertainty Visualization: Applications, Techniques, Software, and Decision Frameworks. IEEE, 2024. S. 62-72.

[6] VASCONCELOS, Francisca; HE, Bobby; TEH, Yee Whye. Uncertainty Quantification in End-to-End Implicit Neural Representations for Medical Imaging.

---

> ### Author Rebuttal · Authors · 2026-03-30
>
> We thank the reviewer for the detailed feedback and valuable references.
>
>
> ## Weaknesses
>
> > W1 & W2: Architecture unclear; improvements from architecture or formulation?
>
> For covariate-conditioned uncertainty quantification, we need ***shapes represented as displacements from a fixed template***, so that each template point corresponds to the same anatomical location across all subjects. This allows comparing subjects in a common coordinate frame and computing spatially varying $\Sigma(\boldsymbol{p},t)$. NAISR and A-SDF are among the few existing methods that model the correspondence between shapes and covariates, which is why we selected them as baselines. However:
>
> - *A-SDF* learns a mapping from spatial coordinates to signed distance values conditioned on a latent code and covariates. It does not produce deformation fields, so there is no spatial correspondence across subjects to define $\Sigma(\boldsymbol{p},t)$.
> - *NAISR* learns target→template deformations conditioned on covariates. Since it does not model the template→target direction, one cannot track a fixed template point across subjects to define per-point $\Sigma(\boldsymbol{p},t)$.
>
>
> We therefore ***decompose the problem into two stages***, following the standard paradigm in statistical shape analysis where registration and statistical modeling are performed separately: (1) a preprocessing stage (Appendix A.1) that establishes template→target correspondences via a hypernetwork and inverse consistency, and (2) PRISM's core model, a standard MLP $f(\cdot): (\boldsymbol{p}, t) \rightarrow (\boldsymbol{\mu}, \Sigma)$. This decomposition makes both tasks more manageable.
>
> We acknowledge that pre-computed correspondences simplify reconstruction. Our experiments aim to show that ***introducing uncertainty quantification and this two-stage decomposition does not degrade performance, and in fact yields improvements*** alongside unique capabilities (local FI uncertainty, local intrinsic time, local OOD detection with AUC=0.832) that are not available in deterministic end-to-end methods. We will add an architectural comparison table in the camera-ready.
>
>
> > W3: Missing related work.
>
> We thank the reviewer for pointing us to CINeMA (Dannecker et al., TMI 2025), CINA (Dannecker et al., MICCAI 2024), SINA (Großbröhmer et al., WBIR 2024), and the uncertainty works by Saklani et al. (2024) and Vasconcelos et al. (2021) — these references help us better contextualize PRISM at the intersection of uncertainty quantification, statistical shape analysis, and implicit neural representations. These works address complementary problem settings (e.g., unconditional atlases, epistemic uncertainty), and a thorough discussion of these connections will strengthen the paper. We will incorporate this in the camera-ready, clarifying how PRISM differs by providing closed-form, covariate-conditioned temporal uncertainty decomposition via Fisher Information while focusing on capturing aleatoric uncertainty.
>
>
> ## Key Questions
>
> > Q1: Architectural and modeling differences.
>
> Please see W1 & W2 above.
>
> > Q2: Role of Eikonal constraints.
>
> Eikonal regularization is used exclusively in the correspondence stage (Eq. 22, Appendix A.1) to ensure valid signed distance fields during template learning. It is not part of PRISM's probabilistic model or uncertainty estimation. Removing it would affect the quality of the pre-computed correspondences but not the PRISM framework itself.
>
> > Q3: Lipschitz regularization.
>
> Thank you for the suggestion. We note that this work (Liu et al., SIGGRAPH 2022\) addresses a different regime: when training shapes are very sparse (e.g., 2 shapes), the network has high epistemic uncertainty at unseen time points, and Lipschitz regularization provides a smoothness prior to prevent artifacts during interpolation. In our setting, the temporal axis is more densely sampled (e.g., 358 scans in the airway dataset), so epistemic uncertainty is less of a concern — the primary challenge is modeling aleatoric population variability. That said, when data is scarce, such smoothness constraints could be beneficial.
>
> One subtlety is that the desired smoothness differs across dimensions: the temporal direction should be smooth, while the spatial direction needs to preserve fine anatomical detail (e.g., sharp transitions between soft tissue and cartilage). Smoothness constraints would need to be applied anisotropically. We consider this an interesting direction for future investigation.

---

> > ### Author Rebuttal · Reviewer_r9XV · 2026-04-01
> >
> > As stated in my initial rebuttal, my concerns mainly revolved around the presentation of the paper, particularly its relation to adjacent work and clarification of architectural considerations. The authors have appropriately clarified my open questions, and I trust the authors to utilize the stated feedback to improve the clarity of the manuscript. Since it is not possible to update the manuscript in the rebuttal, I trust the authors to make the changes in the camera ready. Under this light, I would like to increase my score by *one* point from *weak accept* to *accept*.

---

> > > ### Author Response · Authors · 2026-04-02
> > >
> > > Thank you for acknowledging our rebuttal and for the constructive feedback throughout the review process. We are encouraged by your positive assessment and will incorporate your suggestions to improve the manuscript's clarity in the camera-ready version.

---

### Official Review · Reviewer_725W · 2026-03-13

**Soundness:** 3
**Presentation:** 3
**Significance:** 3
**Originality:** 3
**Overall Recommendation:** 4
**Confidence:** 4

**Summary:**

The paper proposes PRISM, a probabilistic implicit shape model that represents conditional shape evolution with a heteroscedastic Gaussian field, derives a closed-form Fisher-information-based temporal uncertainty measure, and uses an amortized inverse encoder for intrinsic-time inference and downstream tasks such as forecasting and OOD detection. Empirically, the paper evaluates on Starman(G/L), ANNY, and a pediatric airway dataset, with strong results overall, though not uniformly best on every real-data task.

The main takeaway is that the local, uncertainty-aware formulation is the key added value over deterministic baselines such as NAISR and A-SDF, especially for local intrinsic time and airway OOD detection.

**Compliance With Llm Reviewing Policy:**

Affirmed.

**Final Justification:**

I thank the authors for their rebuttal. The additional visualization and the explanation of the design of fisher information address my concern. I have no more questions. I have no more concern and will maintain my original positive scores

**Key Questions For Authors:**

* Can the authors discuss what would be required to scale the framework beyond a single scalar covariate to richer longitudinal settings (e.g., lifetime trajectory of a patient)?

**Limitations:**

Yes

**Strengths And Weaknesses:**

**Strengths**

* Clear methodological contribution: the framework unifies mean trajectory modeling, heteroscedastic uncertainty, intrinsic-time inference, and downstream shape analysis in one model.

* Strong theoretical component: the Fisher information derivation gives a principled motivation for the temporal uncertainty estimator.

* Good experimental scope: synthetic plus clinical data, and multiple tasks including reconstruction, time inference, forecasting, and OOD detection.

* A non-trivial strength is that locality is not only a conceptual claim; it yields a substantial gain for airway OOD detection, where PRISM(Local) clearly outperforms global baselines.

**Weaknesses**

* While the Starman(L) experiments are central to the paper’s claim of modeling local developmental heterogeneity, the empirical analysis is not yet sufficiently diagnostic. In particular, the paper would benefit from clearer side-by-side visualizations showing where global baselines fail and why PRISM’s local formulation is necessary, rather than mainly reporting PRISM’s own local estimates.

* The evaluation does not fully establish robustness on more complex real-world settings. Although PRISM performs strongly on synthetic benchmarks, its advantage is less consistent on the airway dataset; for example, NAISR slightly outperforms PRISM on global intrinsic time estimation in this setting. This makes the current empirical case somewhat less conclusive than the overall narrative suggests.

* Scalability remains insufficiently discussed. The current framework is evaluated with a single scalar covariate, and the manuscript itself acknowledges that extending the approach to high-dimensional covariate spaces is future work. As a result, it is still unclear how well the method would transfer to richer longitudinal settings with more complex temporal or clinical heterogeneity.

* The theoretical analysis is principled, but its practical interpretation should be stated more carefully. The closed-form Fisher information includes both a mean term and a covariance term, yet the practical uncertainty metric focuses on the mean component. An ablation or stronger justification for this design choice would strengthen the theoretical story.

**Minor Issues**

* Visible formatting artifacts such as “AUTHORERR: Missing \icmlcorrespondingauthor.”

---

> ### Author Rebuttal · Authors · 2026-03-30
>
> We thank the reviewer for the thoughtful assessment and the suggestion for stronger diagnostic visualizations and for a scalability discussion.
>
> ## Weaknesses
>
> > W1: Starman(L) empirical analysis not sufficiently diagnostic.
>
> We agree that side-by-side comparisons would strengthen the presentation. We provide a [VISUALIZATION](https://anonymous.4open.science/api/repo/PRISM_rebuttal-87D3/file/PRISM_ICML_rebuttal.pdf) (see Fig.1) comparing shapes generated with local uncertainty (per body part) vs global uncertainty (averaged) on Starman(L), demonstrating where global assumptions fail. We will incorporate this visualization in the camera-ready version.
>
> > W2: Evaluation not fully robust on real-world settings.
>
> We note that on synthetic data, ground-truth intrinsic time is available and PRISM recovers it accurately (Tab. 1). On clinical data, no such ground truth exists — chronological age serves as a proxy. We conducted a test time optimization (TTO) ablation on the airway dataset. PRISM(TTO) outperforms all baselines on time estimation:
>
> | Method | Inference | r ↑ | R² ↑ | MAE (mo.) ↓ | Time/case |
> | :---- | :---- | :---- | :---- | :---- | :---- |
> | NAISR | TTO | 0.908 | 0.820 | 20.512 | 24.36s |
> | PRISM | Amortized | 0.881 | 0.770 | 23.415 | 0.48s |
> | **PRISM** | **TTO** | **0.925** | **0.855** | **18.794** | **8.28s** |
>
> This improvement propagates to personalized prediction:
>
> | Method | Inference | CD ↓ | HD ↓ | EMD ↓ |
> | :---- | :---- | :---- | :---- | :---- |
> | NAISR | TTO | 0.065 | 9.599 | 1.341 |
> | PRISM | Amortized | 0.088 | 12.735 | 1.347 |
> | **PRISM** | **TTO** | **0.064** | **9.031** | **1.264** |
>
> The gap in Tab. 2 stems from the amortized inverse encoder, which provides 17× faster inference (0.48s vs 8.28s) at the cost of a marginal reduction in accuracy. Additionally, PRISM(Local) OOD detection (AUC=0.832) substantially outperforms all global baselines, confirming the clinical value of the local formulation.
>
> > W3: Scalability insufficiently discussed.
>
> The Fisher Information metric generalizes naturally to vector-valued covariates: the scalar $I(\boldsymbol{p},t)$ becomes a Fisher Information Matrix $I(\boldsymbol{p}, t) \in \mathbb{R}^{k\times k}$, where $k$ is the covariate dimension. The diagonal elements quantify the discriminability along each covariate axis independently, while the off-diagonal elements capture the correlations between covariates. The architecture already accepts vector inputs — extending to multi-covariate settings requires no structural changes. We consider this a valuable direction for future investigation.
>
> > W4: Fisher Information mean-only practical interpretation.
>
> The full Fisher Information decomposes as $I_{\text{full}} = I_\mu + I_\Sigma$ (Eq. 9). In Eq. 11 we use only $I_\mu$. This is a deliberate modeling choice: $I_\mu$ and $I_\Sigma$ answer different questions. $I_{\mu} = \boldsymbol{\mu}\_{t}^{\top} \Sigma^{-1} \boldsymbol{\mu}\_t$ measures how precisely we can localize an individual along the developmental trajectory — this is our definition of *temporal variability*. $I_\Sigma$ captures how population diversity itself evolves over time, which is statistically informative but orthogonal to developmental timing. This decomposition follows from the classical orthogonality of mean and covariance parameters under the Fisher-Rao metric (Skovgaard, 1984; Amari, 2016), as discussed in Sec. 4.3 and Appendix A.2.5. Empirically, using $I_\mu$ alone accurately recovers ground-truth uncertainty on synthetic data (Fig. 2), while $I_{\text{full}} = I_\mu + I_\Sigma$ underestimates it (see Fig. 2 in our [SUPPLEMENTARY VISUALIZATION](https://anonymous.4open.science/api/repo/PRISM_rebuttal-87D3/file/PRISM_ICML_rebuttal.pdf)). We thank the reviewer for raising this point and will clarify the justification for focusing on $I_\mu$ in the camera-ready.
>
> > Minor: Formatting.
>
> We will fix all formatting artifacts (AUTHORERR tag, etc.) in the camera-ready version.
>
> ## Key Questions
>
> > Q1: Scaling beyond scalar covariate.
>
> Please see W3 above. For lifetime trajectory modeling specifically, additional covariates (e.g., sex, BMI, disease status) can be incorporated as extra inputs to $f(\cdot)$, with the FI metric capturing their joint influence on shape evolution. The main practical requirement is a sufficiently large longitudinal cohort spanning the age range of interest, and careful treatment of identifiability when covariates are correlated.

---

> > ### Author Rebuttal · Reviewer_725W · 2026-04-02
> >
> > I thank the authors for their rebuttal. The additional visualization and the explanation of the design of fisher information address my concern. I have no more questions.

---

> > > ### Author Response · Authors · 2026-04-02
> > >
> > > Thank you for your thoughtful review and for confirming that our rebuttal addressed your concerns. We appreciate your positive assessment and would be very grateful if your updated evaluation could also be reflected in the score.

---

### Official Review · Reviewer_odnP · 2026-03-13

**Soundness:** 3
**Presentation:** 3
**Significance:** 3
**Originality:** 2
**Overall Recommendation:** 4
**Confidence:** 4

**Summary:**

This paper presents PRISM, a framework that combines neural implicit representations with information geometry to model how anatomical shapes change with covariates like age. The core idea is to learn a heteroscedastic Gaussian field over displacement vectors conditioned on spatial location and time, which simultaneously captures the population mean trajectory and spatially varying uncertainty. A notable theoretical component is a closed form Fisher Information metric that quantifies temporal uncertainty without Monte Carlo sampling. The paper also introduces an amortized inverse encoder for estimating intrinsic developmental time from observed shapes. Experiments are conducted on two synthetic datasets (Starman variants), one semi synthetic dataset (ANNY body model), and one clinical pediatric airway dataset, covering tasks such as shape reconstruction, time inference, personalized prediction, and OOD detection.

**Compliance With Llm Reviewing Policy:**

Affirmed.

**Ethics Expertise Needed:**

["Other Expertise"]

**Final Justification:**

The authors have satisfactorily addressed my concerns. After careful consideration, I have decided to keep my original rating of weak accept.

**Key Questions For Authors:**

What is the exact parameterization of Sigma(p,t)? How many free parameters does the covariance have per query point, and how is positive definiteness enforced during training?

How sensitive is the temporal uncertainty estimate to the warm up schedule and loss weighting? The two stage curriculum (10 epochs L1 only, then joint NLL) seems like a design choice that could significantly affect the learned covariance. Was this ablated?

For the airway OOD detection, PRISM (Global) actually underperforms NAISR (AUC 0.459 vs 0.563). The paper attributes the local version's success to avoiding inter individual confounds, but could this global underperformance indicate that the learned Sigma absorbs too much variance, making global anomaly scores unreliable?

**Limitations:**

See above

**Strengths And Weaknesses:**

Strengths
The problem formulation is clean. Modeling displacement fields as a conditional heteroscedastic Gaussian and deriving a closed form Fisher Information metric is a principled way to get pointwise temporal uncertainty without resorting to sampling. The connection between Fisher Information and the Cramer Rao bound is well motivated and gives the uncertainty a concrete statistical interpretation.

The framework is genuinely unified. A single probabilistic model supports mean trajectory estimation, local/global intrinsic time inference, personalized forecasting, and anomaly detection. This is a meaningful step compared to prior work that typically addresses only one or two of these tasks.

Weaknesses
The covariance matrix Sigma(p,t) is predicted by a neural network, but the paper does not discuss how it is parameterized. Is it a full 3x3 covariance, a diagonal, or a scalar times identity? This matters significantly for the expressiveness of the uncertainty model and for the practical computation of the Fisher Information. If diagonal, the claim of capturing "spatially heteroscedastic" uncertainty is somewhat overstated for correlated deformation components.

The approximation in Eq. (11), treating the Cramer Rao lower bound as an equality, is stated without justification for when this is tight. The bound is only achieved by efficient estimators, and there is no argument that the inverse encoder g or the MLE procedure is efficient. This could lead to systematic underestimation of temporal uncertainty. Some empirical analysis of bound tightness would strengthen the claims.

The comparison baselines are limited. A-SDF and NAISR are both deterministic implicit representation methods. No probabilistic baselines are included (e.g., Gaussian Process Morphable Models, Bayesian neural fields, or even simple ensembles of NAISR). Without comparing against other uncertainty aware methods, it is hard to judge whether the closed form Fisher Information approach offers practical advantages over sampling based alternatives.

---

> ### Author Rebuttal · Authors · 2026-03-30
>
> We thank the reviewer for the insightful feedback.
>
> ## Weaknesses
>
> > W1: $\Sigma(\mathbf{p},t)$ parameterization.
>
> $\Sigma(\boldsymbol{p},t)$ is a full $3\times3$ covariance matrix, parameterized via Cholesky decomposition. The network predicts 6 free parameters per query point (3 diagonal + 3 off-diagonal elements of the lower triangular factor $L$). Positive definiteness is enforced by applying softplus activation to the diagonal elements, and computing $\Sigma=LL^\top$. This parameterization allows modeling of correlated deformation components and is more expressive than the diagonal or $\sigma^2 I$ alternatives. We will add this detail in the camera-ready.
>
>
> > W2: CRB as equality (Eq. 11).
>
> The full Fisher Information decomposes as $I_{\text{full}} = I_\mu + I_\Sigma$ (Eq. 9), corresponding to two orthogonal sub-manifolds under the Fisher-Rao metric (Appendix A.2.5). The CRB uses $I_{\text{full}}$; in Eq. 11 we use only $I_\mu$. This is a deliberate modeling choice, not an approximation error:
>
> $I_\mu$ quantifies *temporal discriminability* — the precision with which the developmental stage can be inferred from local shape observations. $I_\Sigma$ quantifies changes in population dispersion over time, independent of the mean trajectory. By the orthogonality of the mean and covariance submanifolds, they are independent sources of information. Since our goal is to estimate ***the uncertainty of observed shapes projected onto the mean developmental trajectory***, $I_\mu$ is the relevant component, and $1/I_\mu$ is the natural metric for this projected temporal uncertainty.
>
> We provide an ablation comparing $I_\mu$ vs $I_{\text{full}}$ on Starman(G) in our [SUPPLEMENTARY VISUALIZATION](https://anonymous.4open.science/api/repo/PRISM_rebuttal-87D3/file/PRISM_ICML_rebuttal.pdf) (Fig. 2): $I_\mu$ tracks ground-truth uncertainty, while $I_{\text{full}}$ underestimates it.
>
> We also clarify that $I_\mu(\boldsymbol{p},t)$ is ***computed entirely from the forward model*** $f(\cdot)$. The inverse encoder is a ***separate downstream component for fast amortized inference*** and does not enter this computation.
>
>
> > W3: Limited baselines (no probabilistic baselines).
>
> We appreciate this suggestion. Our principle for selecting baselines is that they should model shape conditioned on covariates, and ideally capture aleatoric uncertainty conditioned on covariates (i.e., heteroscedastic $\Sigma(\boldsymbol{p},t)$). However, we are not aware of methods in the implicit shape modeling literature that address this setting. We analyze the suggested methods below:
>
> - *GPMMs* (Lüthi et al., 2017) model unconditional shape variability $p(\boldsymbol{d} \mid \boldsymbol{p})$ with kernel-defined covariance. They do not condition on covariates such as age, and thus cannot capture covariate-dependent temporal uncertainty.
> - *Bayesian neural fields* (e.g., Saad et al., Nat. Comm. 2024, arxiv.org/abs/2403.07657) perform Bayesian inference over network parameters to quantify epistemic uncertainty and not aleatoric uncertainty. Their observation noise is homoscedastic and shared across all spatiotemporal locations — they do not model heteroscedastic covariance conditioned on covariates.
> - *Ensembles of NAISR*: capture epistemic uncertainty (variability from model initialization), not aleatoric population variability conditioned on covariates.
>
> None of these methods model covariate-conditioned heteroscedastic aleatoric uncertainty, which is the central contribution of PRISM. We welcome any additional suggestions and will expand the discussion of these distinctions in the camera-ready.
>
> ## Key Questions
>
> > Q1: Parameterization of $\Sigma(\boldsymbol{p},t)$.
>
> Please see W1 above.
>
>
> > Q2: Warm-up schedule sensitivity.
>
> We ablated the warm-up schedule on the airway dataset:
>
> | Setting | Reconstruction CD ↓ | Time Estimation MAE ↓ | Personalized Prediction CD ↓ |
> | :---- | :---- | :---- | :---- |
> | w/ Warmup | **0.063** | 23.41 | **0.088** |
> | w/o Warmup | 0.064 | **22.18** | 0.098 |
>
> Results are comparable, confirming the model is not overly sensitive to this choice. The warm-up provides a modest but consistent benefit, particularly for personalized prediction.
>
>
> > Q3: PRISM(Global) underperforms NAISR in OOD.
>
> PRISM(Global) uses the global age difference $\hat{\tau}\_{\text{global}} - t$ as the OOD score, following the same protocol as NAISR and A-SDF for fair comparison. Only PRISM(Local) incorporates local geometry and uncertainty normalization (Eq. 17). Two observations: (1) both PRISM(Global) and NAISR perform poorly (AUC 0.459 and 0.563), suggesting that global age-based scoring is not well suited for detecting localized pathology like subglottic stenosis; (2) the same underlying model, when switching to local scoring with uncertainty normalization, achieves AUC=0.832 — a substantial improvement. This demonstrates that local, uncertainty-aware analysis is essential for this clinical task, which is the central motivation of PRISM.

---

> > ### Author Rebuttal · Reviewer_odnP · 2026-04-04
> >
> > na

---

### Decision · Program_Chairs · 2026-04-30

**Decision:**

Accept (regular)

**Comment:**

This submission introduces PRISM, a probabilistic neural representation for modeling covariate-dependent anatomical shape evolution with spatially resolved uncertainty. The method combines implicit neural representations with statistical shape modeling and derives a closed-form Fisher Information metric for local temporal uncertainty estimation. The paper addresses an important problem in medical imaging and presents a unified framework applicable to multiple downstream tasks.

Overall, the paper received consistently positive evaluations, with ratings of one Accept (5) and three Weak Accepts (4). Reviewers broadly agree that the work is technically sound, well-motivated, and makes a meaningful contribution, particularly in integrating uncertainty quantification with implicit neural representations.

In summary, the paper presents a technically sound and conceptually novel contribution and promising empirical results. While some limitations remain in terms of experimental depth and clarity, the positive reviewer consensus supports acceptance.